# Evolution of communication signals and information during species radiation

Maxime Garcia [1,2,10]✉, Frédéric Theunissen [3,4,10], Frédéric Sèbe[1,10], Julien Clavel[5,6], Andrea Ravignani [7], Thibaut Marin-Cudraz[1], Jérôme Fuchs [8] & Nicolas Mathevon [1,9,10]✉

Communicating species identity is a key component of many animal signals. However, whether selection for species recognition systematically increases signal diversity during clade radiation remains debated. Here we show that in woodpecker drumming, a rhythmic signal used during mating and territorial defense, the amount of species identity information encoded remained stable during woodpeckers' radiation. Acoustic analyses and evolutionary reconstructions show interchange among six main drumming types despite strong phylo-genetic contingencies, suggesting evolutionary tinkering of drumming structure within a constrained acoustic space. Playback experiments and quantification of species discriminability demonstrate sufficient signal differentiation to support species recognition in local communities. Finally, we only find character displacement in the rare cases where sympatric species are also closely related. Overall, our results illustrate how historical contingencies and ecological interactions can promote conservatism in signals during a clade radiation without impairing the effectiveness of information transfer relevant to inter-specific discrimination.

[1] Equipe Neuro-Ethologie Sensorielle ENES/CRNL, CNRS, INSERM, University of Lyon/Saint-Etienne, Saint-Étienne, France. [2] Animal Behaviour, Department of Evolutionary Biology and Environmental Studies, University of Zurich, Zürich, Switzerland. [3] Helen Wills Neuroscience Institute, University of California, Berkeley, USA. [4] Department of Psychology and Integrative Biology, University of California, Berkeley, USA. [5] Institut de Biologie de l'École Normale Supérieure, CNRS, INSERM, École Normale Supérieure, Paris Sciences et Lettres Research University, Paris, France. [6] University of Lyon, Université Claude Bernard Lyon 1, CNRS, ENTPE, UMR 5023 LEHNA, F-69622 Villeurbanne, France. [7] Comparative Bioacoustics Group, Max Planck Institute for Psycholinguistics, 6525 XD Nijmegen, The Netherlands. [8] Institut de Systématique, Evolution, Biodiversité ISYEB, Muséum national d'Histoire naturelle, CNRS, Sorbonne Université, EPHE, Paris, France. [9] Institut Universitaire de France, Paris, France. [10]These authors contributed equally: Maxime Garcia, Frédéric Theunissen, Frédéric Sèbe, Nicolas Mathevon. ✉email: maxime.garcia@ymail.com; mathevon@univ-st-etienne.fr

Animal communication signals mediate information flows between individuals and are critical to their survival and reproduction[1,2]. Signals can encode various types of information, including static (e.g. body size, sex, age, identity) and dynamic (e.g. arousal level, physiological states) attributes of the emitter, and are often subject to both sexual and natural selection pressures[3]. As information related to species identity supports mate and competitor recognition, understanding the processes driving signal divergence across lineages has important implications for explaining species' reproductive isolation and interactions within ecological communities[4]. Despite decades of investigation, however, whether signals are under strong selection for species recognition, i.e. whether species-specific information is selected for or against -if at all-, remains an open question[4,5]. Previous studies on signal divergence have emphasized the direct role of natural and social/sexual selection ('sensory drive model')[6–14], the indirect consequences of ecological selection on traits related to signal production ('magic traits')[15–17], as well as the random effects of neutral mechanisms (genetic and cultural drifts)[4,18]. The relative weight of these processes is still debated[4,5,7,18]. For instance, while between-species competition may be a strong driver of signal divergence by promoting niche partitioning[19–21], some studies advocate that competition has little effect[22,23]. Other authors have even reported positive selection for similar signals within multispecies communication networks[24–26]. Similarly, while sexual selection is emphasized as an important driver of signal divergence[13,27,28], it could be of secondary importance compared to neutral genetic drift[22,29–32]. Furthermore, morphological constraints on signal production as well as phylogenetic history may limit evolutionary outcomes and trait diversity in the evolution of acoustic signals[4,18].

An important limitation of our current understanding of divergence mechanisms is that most previous studies have focused on the signal phenotype (e.g. the acoustic features of bird and insect song[11,25] or the colour of body elements[33,34]) without considering the signal functional value, i.e. its actual power to let the receiver decode information, such as species identity[35,36], but see ref. [21]. Recent acoustic playback experiments evidenced that sister pairs of species can display high signal discrimination despite low signal divergence[37–41], emphasizing that structurally similar signals may contain enough information to allow species discrimination. However, these studies focused on a small number of sister species[39,40] or a limited set of signal structural characteristics[37,38,41], restricting our ability to capture the array of evolutionary mechanisms leading to signal divergence and information encoding during a clade radiation. Mathematical tools for quantifying signals' information content have been formalized for a long time (in the framework of Shannon and Weaver's *Mathematical Theory of Communication*[42]), but, to our knowledge, no attempt has been made to investigate how information accompanies signal structural divergence during a clade radiation.

Here we focus on the woodpeckers (Picidae)[43], a family of birds which has developed drumming as an original mode of signalling, to test for selection on signal structural divergence and species-specific information during a clade radiation. Woodpeckers' drumming is a repetitive striking of the beak on a substrate used to communicate species identity in territorial and mating contexts[43–45]. Drumming is an exaptation[46,47], which derived through ritualization (exaggerated amplitude, rhythm stereotypy[48,49]) from pecking on tree trunks, a foraging behaviour typical of woodpeckers[43] (Fig. 1a). A phylogenetic reconstruction of drumming suggests that this was the ancestral behaviour in this family (95% probability of being present in the common ancestor, 22.5 million years ago[50]—see 'Methods' and Fig. 1b). Drumming is an innate behaviour[51], whose divergence has been relatively limited during woodpecker radiation[52], potentially given the strong constraints inherent to its production mechanism[53]. Yet, given their widespread presence within this clade[41] (Fig. 1b) and some evidence of sexual selection on drumming duration and cadence[43,52,54,55], we predicted (1) that signal structure has evolved to at least maintain, and potentially increase, species-specific information during the clade radiation, and (2) that drumming signals provide sufficient information to allow receivers to discriminate conspecifics from sympatric woodpecker species.

To investigate these predictions, our approach is twofold. At the clade level, we rely on phylogenetic-based methods to infer evolutionary patterns of drumming structure and information content. However, encoding of species-identity information is not biologically relevant at the clade level, since not all species occurred in the same place and time. Thus, we also narrow down our approach to the ecological community level, where we can investigate the fine mechanisms through which species discrimination operates. We first quantify drumming acoustic structure and species-specific information in a large sample of extant species. Second, we build evolutionary reconstructions coupled with information-theoretic simulations of signal discriminability[56] to test whether the emergence of novel drumming types during woodpecker radiation has led to increased information. Third, we conduct field behavioural experiments and examine species assemblages at the level of ecological communities to evaluate the actual species discriminative power of woodpeckers' drums in relation to sympatry. Overall, we show that within a relatively limited acoustic space, the evolution of drumming signal allows efficient species discrimination, despite a lack of strong selection pressure to increase species-specific information content. In particular, adjustments made within species assemblages are key for such mechanisms to unfold. This study on signal evolution at both clade and community levels encompasses the whole communication chain, from signal production by the emitters to information decoding by the receivers.

## Results

**Information-theoretic estimations.** We quantified the amount of information supporting species discrimination by characterizing the acoustic structure of drumming signals for 92 species of woodpeckers, based on 22 acoustic variables (see 'Methods', Supplementary Table 1). Given the variation in acoustic structure of drumming patterns across the woodpecker family, we predicted that a bird species could be identified based on its drumming alone and used information theory to quantify the upper limit of one's performance in this species discrimination task. Using a hierarchical clustering analysis, we distinguished six drumming types that are well segregated in a drumming-specific acoustic space (Fig. 2a, Supplementary Fig. 1) and could be characterized accurately (see Methods 'Acoustic data and analysis' section for definitions). We then performed a discriminant function analysis (DFA) to generate a species classification matrix ('Methods'; Fig. 2b) and used the Shannon's Entropy equation[42] to calculate the local mutual information ($MI_L$) values characterizing the information content of each species' drumming signal (see 'Calculation of information' in 'Methods'; Supplementary Fig. 2a). The average rate of correct classification was 16.5%, significantly above what is expected by chance (chance = 1/number of species = 1.09%; permuted DFA: $P < 0.001$, Supplementary Fig. 3). This result both indicates that drumming does contain species-specific information (in line with field results showing species discrimination among woodpecker species[45]), and yet that within the woodpeckers' clade, a randomly chosen

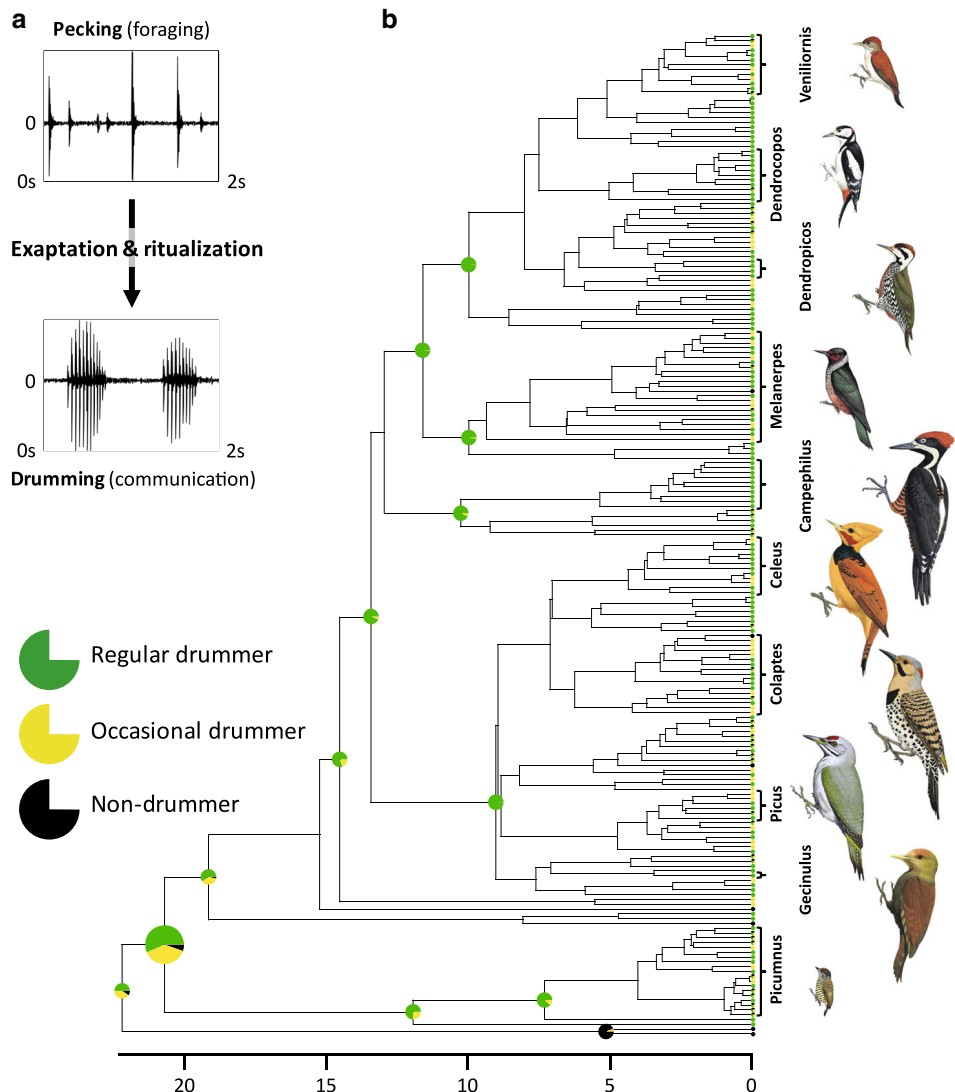

**Fig. 1 Origin of drumming in woodpeckers. a** Pecking, a foraging behaviour used in woodpeckers to excavate prey from tree trunk, has acquired a stereotypical and rhythmic structure, thus exapting into drumming, which holds a communicative function during mating and territorial defence. **b** Time-calibrated phylogenetic tree based on molecular data obtained by Bayesian inference. The tree shows the actual and the derived ancestral drumming behaviour in the woodpecker phylum. Pie charts are empirical Bayes posterior probabilities of ancestral states on internal nodes obtained from a continuous time Markov model (see 'Methods'). For clarity, only selected pie charts are displayed at the nodes of large clades. Written genera (10 shown) indicate some of the groups included in subsequent analyses (total: genera: $n = 22$; species: $n = 92$; see Supplementary Data 1). Age is indicated in million years (Illustrations of woodpeckers reproduced by permission of Lynx Edicions). Source data are provided as a Source data file.

drum can have a high probability of being attributed to a wrong species. Misclassifications, however, are far from random (Fig. 2b) with most errors in classification occurring between closely related species. These systematic misclassifications can be used to further deduce the potential species identity. For example, particular misclassifications could be determined to be implausible, or even impossible, because the misclassified species and the correct species are clearly distinguishable based on visual cues, or because the misclassified species is never found in the location where the drum is heard. In such scenarios, after eliminating the most probable (but incorrect) species, the second most probable species based on acoustic features of the drum heard can be the correct species. The measure of overall mutual information (MI) captures not only the probability of correct classification found in the diagonal of the classification matrix but also the potential information that is found in the systematic misclassifications. MI is expressed in bits. One bit means that woodpecker species could be perfectly classified into two groups based on their drum; 2 bits

means that species could be perfectly classified into 4 groups; and so forth. Since we analysed the drum of 92 species, the maximum number of groups that could be perfectly discriminated, with one species in each group, is 92. Thus, the maximum MI achievable is log2(number of species) = 6.52 bits. In our analyses, we used a normalized MI by dividing the MI by this ceiling value since we will later compare MI calculated for variable number of species. The normalized overall MI was 38% ± 19% (corresponding to 2.48 ± 1.24 bits; see Supplementary Fig. 2a and 'Methods' for additional details). To gain further intuition on the magnitude of this information-theoretic measure, the MI can be translated into the correct classification that would be achieved if misclassifications were indeed random (see 'Methods', Eq. (3)). If misclassifications were random, reaching the observed MI of 2.48 bits would require 53% correct classification. Thus, information for species identity is clearly present in the drumming signal and could provide reliable species discrimination as long as particular errors of classifications within related species could be avoided.

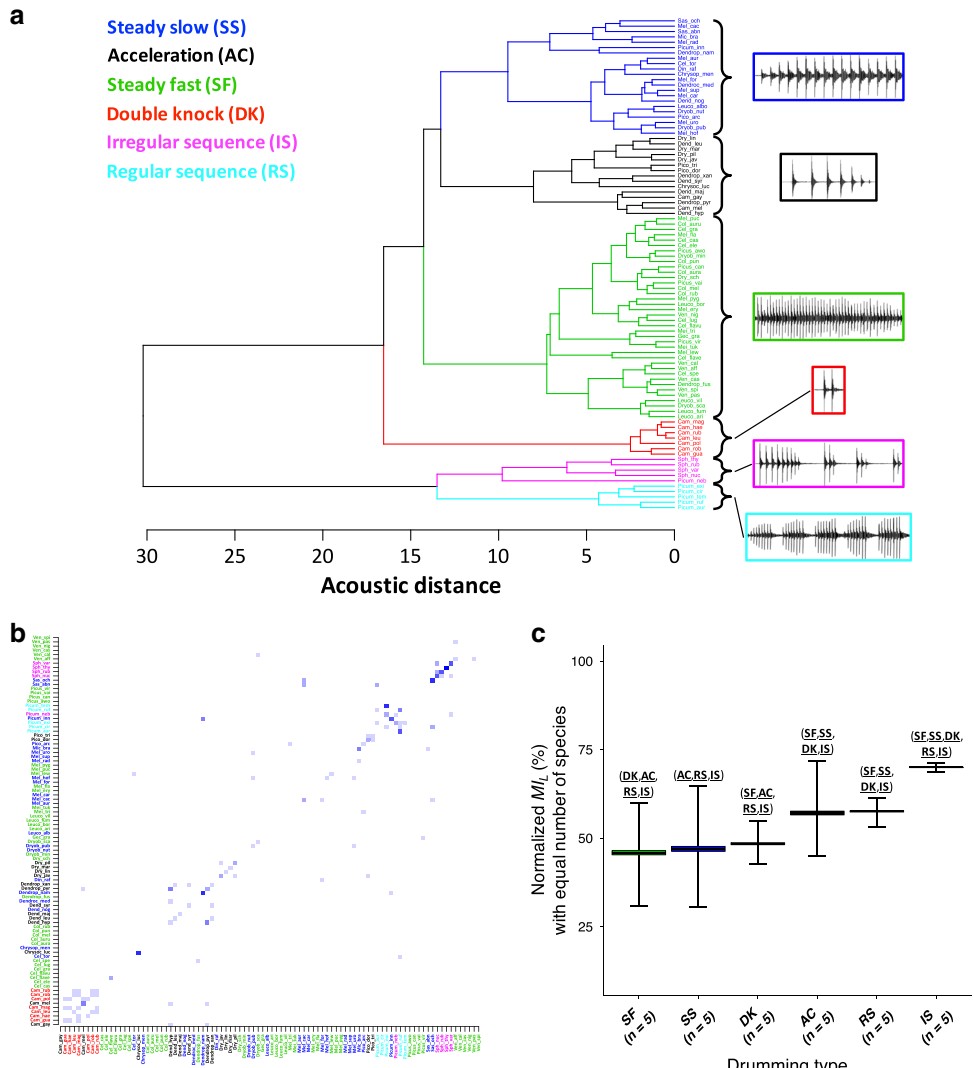

**Fig. 2 Quantification of signal structure and information for species identity. a** Hierarchical clustering of drumming signals based on acoustic Euclidean distances. Oscillograms illustrate the six drumming types: 'steady slow', 'acceleration', 'steady fast', 'double knock', 'irregular sequences' and 'regular sequences' (analysis based on 22 acoustic variables; $n = 92$ woodpecker species; 3–39 drums/species; see Methods 'Acoustic data and analysis' section). **b** Confusion Matrix showing the conditional probability of classification (given a specific species) resulting from a discriminant function analysis (DFA) using the 6 principal components summarizing drumming acoustic structure. Rows ($Y$ axis) are actual species and columns ($X$-axis) correspond to the species predicted by the DFA. Each row shows the conditional probabilities of predicting a specific species shown on the $X$-axis given the species on the $Y$ axis. This probability is written as $p(X_M|X_A)$ in the 'Methods' section. These probabilities are colour-coded on a blue scale, the darker the blue the higher the posterior probability (and thus the better the classification). Perfect classification would appear as a dark blue diagonal. Note the localized increase in misclassification occurring among closely related species (illustrated by 'clusters' of blue squares: species are ordered alphabetically, thus species from the same genus are adjacent on the x and y axes). **c** Normalized local mutual information ($MI_L$) for each drumming type when considering equal number of species per type (100 iterations; box plots denote mean ± SE (boxes) and min/max values (whiskers); species were randomly selected for each iteration among the species available within their drumming type). Tests and significance levels are indicated as follows: (SF) significant difference with SF; (SS) significant difference with SS; (AC) significant difference with AC; (DK) significant difference with DK; (RS) significant difference with RS; (IS) significant difference with IS. Abbreviations without formatting indicate $P < 0.05$; abbreviations in bold indicate $P < 0.005$; abbreviations in bold and underscored indicate $P < 0.001$; $P$ values are adjusted for multiple comparisons between groups (one-way ANOVA, Tukey post hoc test; two-sided statistics are reported). Source data are provided as a Source data file.

We analyze this putative strategy in the section 'Information in ecological communities' below.

Since our hierarchical clustering analysis of acoustic structure revealed a finite number of distinct drumming strategies, we began to explore the relationship between acoustic structure and species signature by examining the contribution of drumming type to the *MI* for species discrimination. We postulated that novel (in the sense 'newly emerged') drumming types might evolve to increase the *MI* during clade radiation, and examined this hypothesis based on the 92 extant species in our dataset. It appears that some drumming types lead to higher information values, e.g. 'irregular sequence' (IS) encodes species-specific information significantly better than other types when controlling for the number of species using each type (Fig. 2c; see Methods 'Calculation of information' section for details). An IS drumming type is more informative because it is more different than other types from an 'average' drumming type (i.e. the distribution of drumming acoustic features obtained for all species). However, as

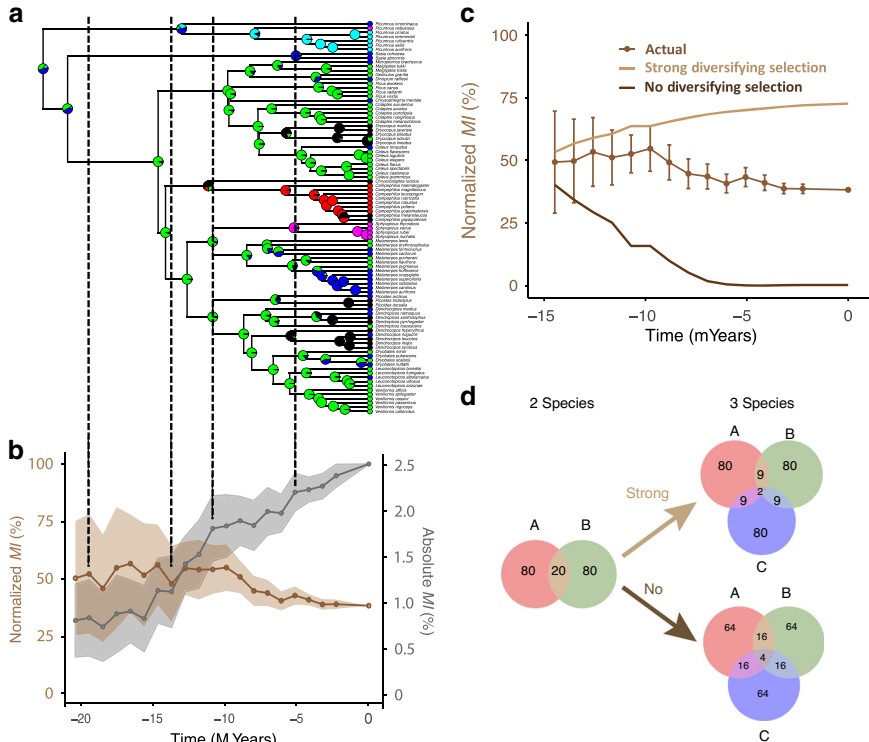

**Fig. 3 Evolution of drumming: acoustic structure and information. a** Ancestral state reconstruction of drumming types along the woodpeckers' phylogenetic tree. Pie charts at nodes represent the probability distribution of existing (colour-coded as in Fig. 2a) drumming types. **b** simulated information content associated with reconstructed drumming types; normalized overall mutual information (*MI*; brown curve) is scaled on the left *y* axis, while absolute *MI* (grey curve) is scaled on the right *y* axis—see 'Methods' for details. Shaded error bars show ±1 SD of mean information values obtained in simulations. **c** Ancestral normalized *MI* (as in **b**) compared to two modelled calculations corresponding to strong (light brown) and no (dark brown) selection pressure for species identity—models included $n = 30$ simulations and error bars show ±1 SD of mean information values obtained in simulations. See 'Analytical simulations of selection for information' and Supplementary Fig. 6 for details). **d** Venn diagrams illustrating the assumptions used to simulate evolutionary scenarios with strong or no diversifying selection in (**c**). Numbers indicate theoretical percent probability of correct classification (e.g. with a strong diversifying selection, the correct classification probability remains at 80% for all species when a new species emerges, but drops to 64% when no diversifying selection is applied). Source data are provided as a Source data file.

expected from increased similarity in drumming structure, overall the amount of species-specific information decreases with the number of species sharing the same drumming type (Supplementary Fig. 2b; Pearson's $r = -0.89$, $P = 0.02$). Therefore, the increase of information provided by the onset of an emerging drumming type within a woodpecker lineage is likely to be "washed out" by the species radiation that follows. This 'wash-out' pattern should be at its strongest when no direct acoustic competition occurs between species, because closely related species are more likely to share a similar signal (as indicated by the strong phylogenetic signal found on drumming acoustic structure—Fig. 1b, Supplementary Figs. 4, 5 and Supplementary Table 2).

**Evolutionary reconstructions.** Considering the strong historical contingencies characterizing the evolution of drumming signals (Fig. 1b; Supplementary Figs. 4 and 5, Supplementary Table 2), we then evaluated changes in the amount of species-specific information as new drumming types evolve and new species appear. To do so, we produced evolutionary reconstructions of drumming types and of their associated information content (information-through-time plots; Fig. 3a) along the woodpeckers' phylogenetic tree (see 'Methods'), predicting that signal structure should have evolved to optimize species-specific information during the clade radiation. This analysis showed a steady increase of information during species radiation, as expected if the between-species versus within-species variance of the drumming

signal remains constant and the number of species (or signals to be encoded) increases (Fig. 3b, grey curve). To better assess how information-through-time changes, we normalized it by its ceiling value (i.e. by the maximum amount of information that can be encoded while discriminating $n$ species). Remarkably, this normalized species-specific information remained relatively constant during the woodpecker radiation (Fig. 3b, brown curve), highlighting how selection pressures acted to maintain species discrimination even as the number of species increased along our phylogenetic reconstruction. In comparison, analytical models show that a completely random evolution of drumming signals would have led to a drop of normalized information as the number of species increased (Fig. 3c, dark brown curve). At the other extreme, strong evolutionary pressures only allowing for new species whose drumming signal would yield the same probability of correct detection would have led to a significant rise in normalized information as the number of reconstructed lineages increased (Fig. 3c, light brown curve; see Fig. 3d and 'Methods' for assumptions in the analytical simulations). The reconstructed actual evolutionary trajectory of normalized species-specific information is between these two extremes (Fig. 3c, brown curve). A finer examination of the dynamics of the information-through-time plot shows more rapid progression of absolute information as new drumming types appeared along woodpecker radiation (Fig. 3a, b, dotted lines). Indeed, the observed species-specific information showed a larger increase than when simulated without the onset of new drumming types

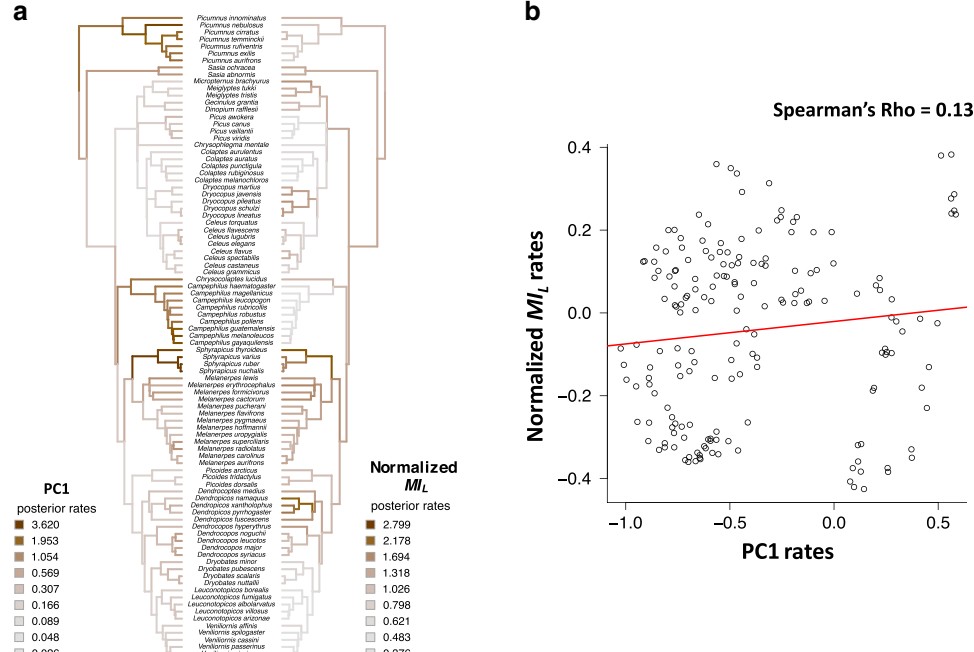

**Fig. 4 Evolutionary changes of drumming structure and species-specific information. a** Reconstruction of the evolutionary rates for drumming acoustic structure ('PC1') and normalized $MI_L$ along the woodpeckers' phylogeny ($n = 92$ species). Both trees are based on relaxed-rates Brownian motion evolutionary models. Average posterior rates have been standardized (dividing variables by their respective standard deviation before model computation) and are represented by a colour-gradient (scales indicated in the Figure). These gradients do not indicate an increase or decrease of the absolute trait values, but rather whether the rate of change (i.e. the diversification speed) of a trait increases or decreases. **b** Correlations between evolutionary rates of acoustic structure (represented by PC1) and $MI_L$. Rates were log-transformed; for acoustic structure represented by PC2–PC6, see Supplementary Fig 10. Source data are provided as a Source data file.

(Supplementary Fig. 6). This suggests that the emergence of new drumming types, and the resulting increase in information at the clade level, was balanced out by a growing number of species to discriminate. Thus, drumming signals do not appear to have been under high evolutionary pressure to maximize information for species identity, nor did they randomly drift as this would have resulted in a decrease in information. Instead, selective pressure acted to preserve the amount of species-specific information that was generated when drumming first appeared in woodpeckers. The evolution of novel drumming types also reflects this balanced course: the transition probability matrix resulting from the reconstructed evolutionary history of drumming suggests that drumming types did not appear in an order which would have increased their amount of species-specific information (Supplementary Fig. 7). Instead, drumming types interchanged during woodpeckers' radiation, with fluctuations suggesting an evolutionary tinkering of drumming structure within a constrained acoustic space.

We further tested if evolutionary changes in the acoustic structure of drumming signals correlate with changes in species-specific information using a Bayesian approach under a relaxed Brownian motion model (see Methods 'Ancestral states reconstructions' section for details). The evolutionary rates of change in acoustic structure and species-specific information significantly differed along the clade's phylogeny (Fig. 4a; Supplementary Figs. 8 and 9; Supplementary Table 7) but still showed moderate correlations ($0.04 <$ Spearman's $Rho < 0.41$; Fig. 4b; Supplementary Fig. 10; Supplementary Table 7). For instance, while major structural changes occur in clades producing IS (*Sphyrapicus*), RS (*Picumnus*) and DK (*Campephilus*) drumming types (Fig. 4a), the associated $MI_L$ only increased for the IS drumming type. This synchronous change for IS is consistent with our findings showing that this

drumming type seems to be the most informative (Fig. 2c). Furthermore, producing SS and SF drumming types is characterized by increased variation in the rates of $MI_L$ (Fig. 4a), which corresponds to clades (e.g. *Melanerpes*) where transitions between structures are more frequent (Fig. 3a). These results highlight a partial decoupling between drumming structure and function, and emphasize that the magnitude of evolutionary changes differs between these traits, with structure overall undergoing faster changes than species-specific information throughout the woodpecker radiation.

Finally, we investigated the current association between drumming structure, species-specific information and life-history traits using phylogenetic generalized least square regressions (see Methods 'Phylogenetic generalized least squares' section for details). Two key findings can summarize this approach. First, we showed (in line with the partial decoupling described above) a significant correlation between signal acoustic features (PC1) and species-specific information (Supplementary Tables 3 and 4). This indicates that drumming's species-identity content is particularly well encoded by the variables strongly ($\geq 0.7$ or $\leq -0.7$) loaded on PC1: drum duration, temporal and amplitude jitter, variables related to the maximum time interval between two consecutive pulses, acceleration, and number of pulse sequences per drum (Supplementary Tables 1, 11 and 12). In other words, currently, drums that are longer, with higher amplitude and temporal jitter, a longer maximum inter-pulse interval (as long as this interval does not appear in first or last position), which accelerate and are displayed as sequences (bouts) convey more information about species identity. Second, and conversely, morphometric and geographical distribution variables (typically affecting signals' acoustic structure[4]) were poor predictors of both signal structure and species information (Supplementary Tables 3–6). Only drumming peak frequency

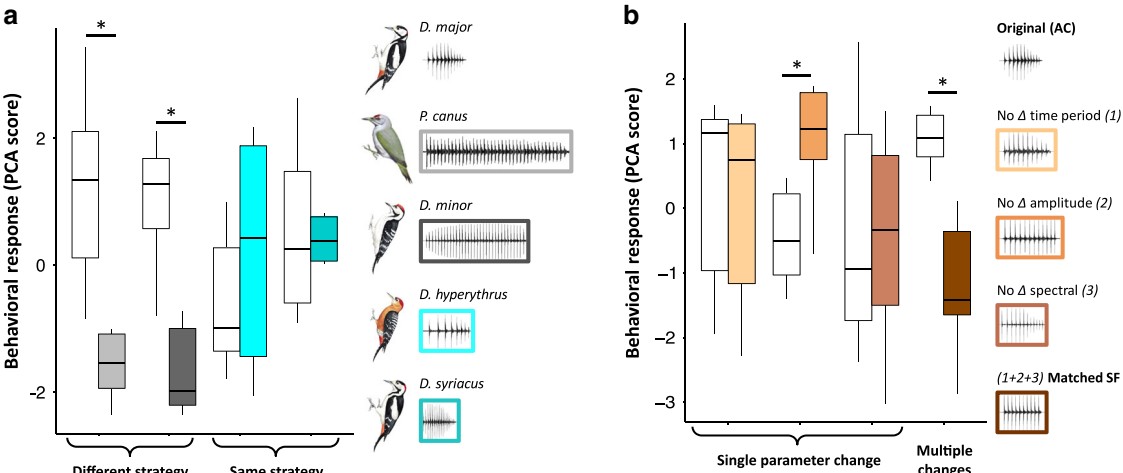

**Fig. 5 Field playback experiments: behavioural response of *Dendrocopos major* to played back signals. a** Compared with presentation of drums from their own species (white box plots), individuals (*n* = 6 for each condition) did not respond to species with a different drumming type (*Picus canus* (*P* = 0.001; paired comparison using linear mixed model (LMM) and LSmeans contrasts) and *Dryobates minor* (*P* = 0.009; paired comparison using LMM and LSmeans contrasts)) whereas they reacted strongly to species sharing the same type (*Dendrocopos hyperythrus* and *D. syriacus*; both *P* > 0.05; paired comparison using LMM and LSmeans contrasts). **b** Similar to panel a, we tested *n* = 6 individuals in each condition. Individuals were tolerant (no decrease in response) when only one acoustic dimension of the original signal was changed (suppression of either the inter-pulse time variation in the drumming sequence, the amplitude modulation, or the spectral modulation between successive pulses). Birds' reactions significantly dropped when all acoustic dimensions were affected simultaneously ('multiple changes' condition; *P* < 0.001; paired comparison using LMM and LSmeans contrasts). The higher the behavioural score, the stronger the response (i.e. birds reacted faster, approached closer and with a stronger acoustic response to the playback sequence). In both panels, box plots denote median and 25th to 75th percentiles (boxes) and min/max values (whiskers) (illustrations of woodpeckers reproduced by permission of Lynx Edicions). *Significant difference between conditions using pairwise comparisons and Tukey adjustment for multiple testing; two-tailed statistics are reported. Source data are provided as a Source data file.

(approximated via PC6—see Supplementary Table 11) and species body size (approximated via wing length and body mass) were found to significantly correlate (Supplementary Tables 4 and 5): bigger birds drum with lower dominant frequencies. While isolated, this observation has strong potential value: it most likely reflects the fact that bigger birds hang in and drum on larger trees, which have lower resonance properties. This association between anatomy and drumming structure could indicate a key influence of habitat type on species spatial distribution, which in turn could affect species discrimination requirements within communities.

Linking back to our initial predictions, these models overall highlight that structural changes in drumming structure were accompanied by the maintenance, rather than the increase, of species-specific information during the clade's evolutionary history. This evolutionary pattern seems to have unfolded mostly regardless of life-history traits at the clade level but highlights the need and relevance of complementary analyses at the level of ecological communities.

**Field behavioural experiments**. To evaluate the biological relevance of species-specific information found in drumming signals at the clade level, we conducted field playback experiments on the great-spotted woodpecker *Dendrocopos major*. In particular, we tested the birds' ability to discriminate between conspecific and heterospecific signals in natural conditions. *D. major* is a European species whose drum falls in the 'Acceleration' type (Fig. 2a). In the first set of experiments, we tested the actual relevance of drumming types by assessing the behavioural reaction of *D. major* to conspecific versus heterospecific drums. Tested individuals responded more strongly to drums of their own species, than to that of two sympatric species with a different ('Steady Fast') drumming type (linear mixed model (LMM), *Dryobates minor*: $\beta = -1.07$, $t = -2.88$, $P = 0.009$; *Picus canus*:

$\beta = -1.40$, $t = -3.76$, $P = 0.001$; Fig. 5a). Conversely, they responded as strongly to drums from two other non-sympatric species sharing the 'Acceleration' type (*Dendrocopos syriacus*: $\beta = -0.11$, $t = -0.31$, $P = 0.76$; *Dendrocopos hyperythrus*: $\beta = 0.40$, $t = 1.01$, $P = 0.29$; Fig. 5a) as to drums from their own species, showing that they lacked species information in this context. In a second set of experiments, we tested the level of tolerance of *D. major* to alterations of conspecific drums by presenting individuals with synthetic drums with modified acoustic features. These modifications affected various aspects of the acceleration pattern, i.e. either the temporal variation (by imposing a steady pulse rate), the amplitude variation (by normalizing the amplitude of pulses), the spectral variation (by normalizing the spectral properties of pulses), or these three parameters simultaneously. Tested individuals failed to discriminate the first three modified signals from natural conspecific drums (temporal variation: $\beta = -0.13$, $t = -0.59$, $P = 0.56$; spectral variation; $\beta = -0.09$, $t = -0.41$, $P = 0.69$; even responding with higher intensity to the modification of amplitude variation, likely because the overall louder amplitude would simulate a much closer intruder and thus represent a superstimulus[57]: $\beta = 0.74$, $t = 3.22$, $P = 0.004$; Fig. 5b). However, their behavioural response decreased significantly when all three parameters were simultaneously altered ($\beta = -1.11$, $t = -4.96$, $P < 0.001$; Fig. 5b). Thus, only a severe modification of the drumming structure leads to changes in the signal informative content. These field experiments further support the lack of a strong selective pressure to increase species-specific information in this communication system at the clade level, while showing that species recognition of sympatric heterospecifics is effective.

**Information in ecological communities**. We then investigated further the hypothesis that woodpeckers living in sympatry evolve distinguishable drumming types, predicting character displacement

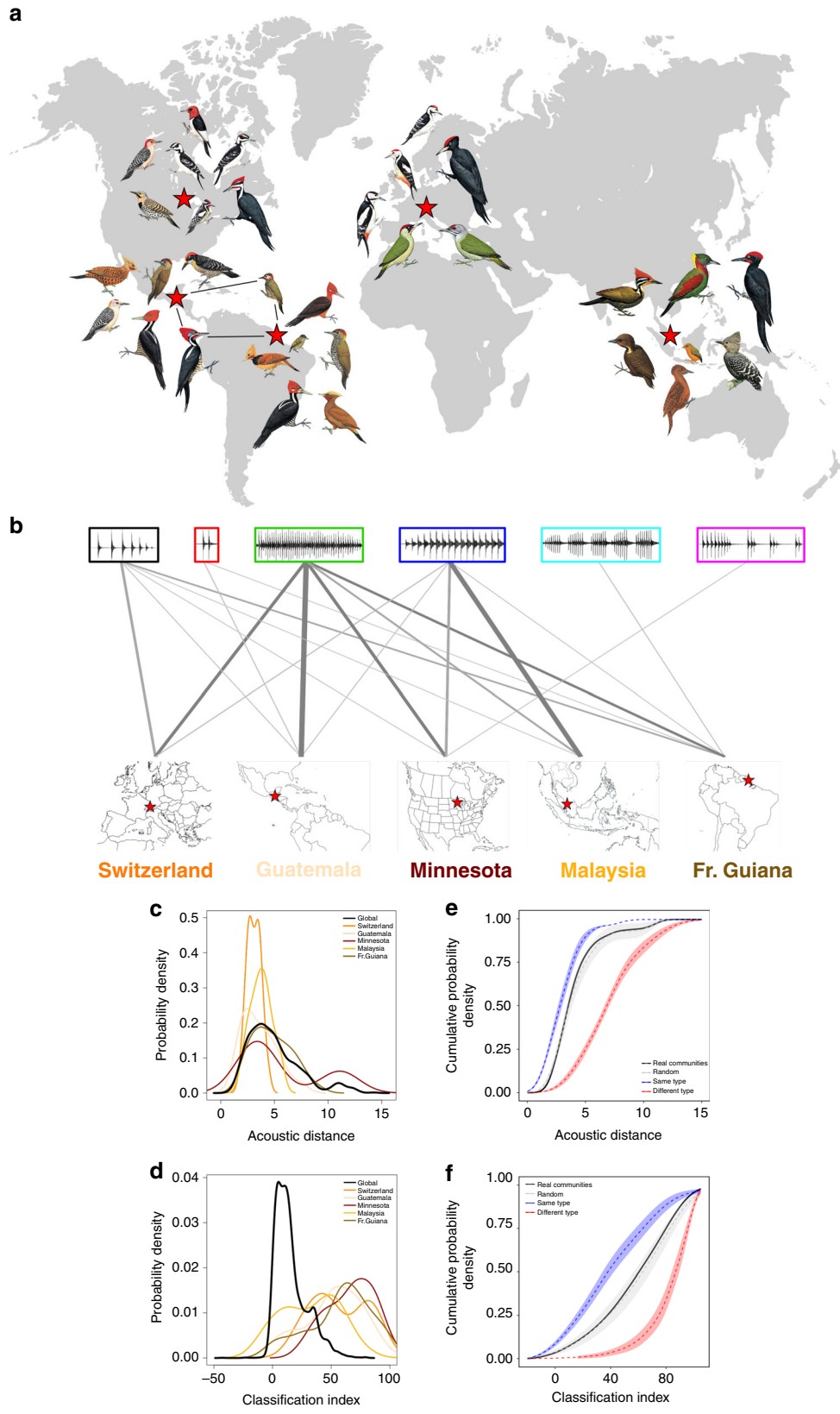

in cases where sympatric species shared a similar enough drumming structure. According to published data from various areas around the globe (Fig. 6a; Supplementary Table 8), woodpeckers are typically organized in ecological communities of six to ten sympatric drumming species. Within a community, sympatric species distribute their displays among three to five different drumming types among the six types described above (Fig. 6b). Moreover, the distributions of pairwise acoustic distances between species drums are similar within communities and in comparison to the overall distribution in the clade (Fig. 6c). Thus, the usage of drumming acoustic space is mostly similar within communities and the entire *Picidae* clade (see also Supplementary Table 9). We

**Fig. 6 Drumming within ecological communities. a** Distribution and species composition of the communities examined in this study (background map designed by Layerace/Freepik). **b** Diversity of drumming types used by sympatric species in ecological communities (from left to right: palaearctic deciduous forest, Switzerland; neotropical forest, Guatemala; nearctic deciduous forest, USA; Indomalayan rainforest, Malaysia; neotropical rainforest, French Guyana; Supplementary Table 8). Line width is proportional to the number of species using a given drumming type. **c** Average acoustic distance between species pairs either taking all species together (black line, $n = 92$) or within communities (coloured lines). Acoustic distances are computed as the Euclidean distance between the 22-dimension vectors for each species' drum. The average acoustic distance mainly shows similar density profiles for the woodpecker family taken as a whole and for species living in sympatry (i.e. within communities), showing an optimal use of acoustic strategies within communities (Supplementary Table 9). **d** Average Classification Index within species pairs, either taking all species together (black line; $n = 92$), or within communities (coloured lines). Classification Index ranges from −100 (no correct classification at all) to 100 (maximum correct classification). Compared with the family taken as a whole, sympatric species (i.e. within communities) are much better classified (Supplementary Table 9). **e** Cumulative distribution of the average acoustic distance within species pairs either taking distances within communities ('Real Communities'; black line), or randomly sampling species from our full community dataset (i.e. 6 species among 33; grey line), or selecting species each with different drumming types (red line), or selecting species which all have a similar drumming type (blue line). The sooner the cumulative distribution reaches a plateau, the lower the average acoustic distances between species pairs in a simulated community category. **f** Cumulative distribution of the average Classification Index within species pairs, following the same approach as with acoustic distances (note that computing communities with similar drumming types was limited to 5 species when sampling from the 'regular sequence' and/or from the 'irregular sequence'). The later the cumulative distribution reaches a maximum, the higher the percent correct classification in a simulated community category. For panels (**e**) and (**f**), a balanced selection of 5 and 6 species per community was applied to control for a possible effect of the number of species; error envelopes were obtained by bootstrapping ($n = 100$, curves are mean ± 2 SD) (illustrations of woodpeckers reproduced by permission of Lynx Edicions). Source data are provided as a Source data file.

also compared the resulting pairwise discrimination performance by calculating a Classification Index (CI) (see Methods 'Calculation of information' section). The pairwise correct classification is higher in communities than when it is estimated for all species in the clade (Fig. 6d; Supplementary Table 9). However, this gain in discrimination could be both due to the smaller number of species within a community (making the classification task easier) or to community-specific features in the drums that could facilitate classification. To disentangle these two effects, we simulated virtual communities all with equal number of species (5 and 6) and either composed with random species or with species all having a similar or a different drumming type. Those simulations clearly showed the advantage of all species having distinct drumming types over all species having similar types within a community (red vs blue dashed lines in Fig. 6e, f). The simulations also showed that the actual communities (black lines) had performances that were between these two extremes and similar to a random sampling (grey lines) of species (Fig. 6e, f). In summary, these results and simulations suggest altogether that acoustic discriminability is facilitated by the low number of species constituting a given community as well as by the use of distinct drumming types within that community. Woodpecker species within a community do not need, however, to separate their drums acoustically to the furthest extent possible, as long as their drumming types are as different as those found in the entire clade.

It remains to be seen whether such random distribution of species within communities is expected. Character differences are predicted to be accentuated in species with overlapping geographic distributions compared to species that do not co-occur in the same areas as a result of competitive exclusion: a phenomenon known as character displacement[4,58]. Through this process, we could expect phenotypic (in this case, drumming structure) differences to be as high for closely related species that share the same geographical area as for distantly related species or allopatric species. This may ultimately lead to some cases of sympatric speciation within woodpeckers, although such mechanism appears to be relatively rare among birds[59]. To examine potential character displacement in woodpeckers' drumming, we correlated pairwise phylogenetic distances with acoustic distances and classification for sympatric and non-sympatric species. By considering all possible pairs of species in our sample ($n = 4186$ pairs, with 598 pairs identified as sympatric; 92 species), we found that character displacement of drumming structure occurs where sympatric

species are closely phylogenetically related (Fig. 7a; Supplementary Table 10). This confirms the inter-specific discriminative function of drumming signals, and suggests that the divergence in signals is reinforced by the positive consequence of potentially being correctly classified (Supplementary Fig. 12), as shown by increased rates of classification within sympatric species pairs (as opposed to non-sympatric pairs) when phylogenetic distances are low (Fig. 7b; Supplementary Table 10). In contrast, the effect of sympatry becomes negligible as species pair are more distantly related (Fig. 7a; Supplementary Table 10). Indeed, in line with the observation that drumming structure is highly conserved within a clade (Supplementary Figs. 4, 5, and Supplementary Table 2), when sympatry involves species pairs that are distantly related, classification is significantly better than when it involves closely related species (Fig. 7b; Supplementary Table 10). Remarkably, only 70 out of the 598 sympatric species pairs involve two species from the same genus. This low relatedness among sympatric species may indicate that the diversification process in woodpeckers is mainly allopatric, and followed by secondary speciation[60]. Alternatively, it could be a consequence of foraging competition that excludes close species occupying the same ecological niche[61,62] (note that this mechanism would apply equally for closely and distantly related species). Combined with our results showing a significant association between species' body size and drumming peak frequency (see results from our PGLS analyses), these patterns overall suggest that the refining of species-identity encoding processes is only moderately affected by character displacement of drumming structure, but more likely dependent on ecological factors such as species distribution, ecological resources and/or habitat type.

## Discussion
Our study provides the first evidence that limited signal divergence at the scale of a clade radiation does not impair discrimination between sympatric species found within communities. By reconstructing the evolutionary history of signal's information content in parallel to signal structure, our work adds and quantifies a functional perspective to evolutionary patterns, thereby offering novel insights into animal signal evolution. Phylogenetic analyses allowed us to establish the broad-scale evolutionary patterns found within a clade radiation. This step is key for investigating how a signal's acoustic space has been explored in a particular clade to represent species information, as

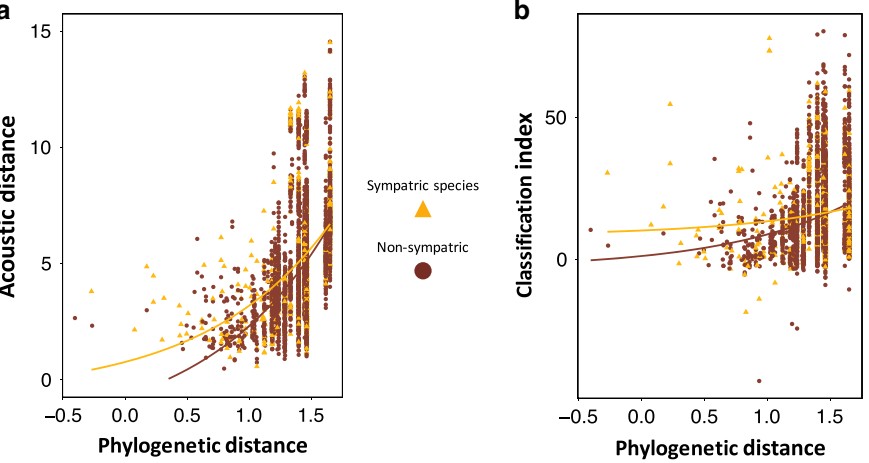

**Fig. 7 Effect of sympatry on the relationship between drumming acoustic structure and species-specific information content. a** Relationship between acoustic (Euclidean) and phylogenetic (million years, log-transformed) distances between species. Each dot represents a pair of species. Acoustic distances are computed as the Euclidean distance between 22-dimension vectors (for 22 drumming acoustic features; one vector per species). Within panel (**a**): closely related species have more similar signals, which are more likely to be affected by sympatry; i.e. character displacement due to sympatry only occurs for low phylogenetic distances, see statistics in Supplementary Table 10. **b** Relationship between the Classification Index (ranging from −100 to 100) and phylogenetic distance (log-transformed) between species. Each dot represents a pair of species. Within panel (**b**): signals from closely related species are more likely to be affected by sympatry to increase potential classification between species; i.e. increase of correct classification due to sympatry (through the character displacement observed in panel **a**) only occurs for low phylogenetic distances, see statistics in Supplementary Table 10. Source data are provided as a Source data file.

well as for making predictions about the actual discrimination processes occurring at a biologically relevant scale. Using our understanding of drumming signal structure and species discrimination potential at the clade level, our playback experiments and ecological analyses pinpointed the actual mechanisms that ensure efficient species-identity encoding within communities. We show that historical contingencies leading to the strong phylogenetic signal on drumming behaviour are balanced by ecological arrangements. Rearrangement of species composition and/or spatial distribution (which here corresponds to an increased phylogenetic diversity) offer an optimal distribution of drumming types within this signal's acoustic space. This process seems complemented by punctual character displacement of drumming acoustic structure in cases where sympatric species are also closely related, overall showing how species discrimination operates in natura.

Overall, our results suggest that woodpeckers' drumming signal has not been selected for maximizing information for species identity in the signal itself but that it has nonetheless preserved the efficacy of its species' signature as the number of species in the clade increased. Thus, while drift[4,63] may represent a global underlying process in the evolution of drumming, other forces have locally shaped this communication signal further and allowed the observed partial decoupling between acoustic structure and information content. In this context, acoustic cues to species identity may have emerged as a by-product of other processes (considering the existing but limited effect of character displacement we observed). We suggest that foraging competition[61,62] is likely the major factor relaxing the selection pressure for a higher level of species discriminability in drumming at the community level. Foraging niche exclusion between closely related species (which we have shown to be more likely to share similar drumming types—Supplementary Fig. 4) would indeed relax selective pressures towards signal divergence and therefore facilitate the maintenance of drumming's strong phylogenetic signal at the clade level. A possible outcome of reducing ecological interactions between different species (in this case foraging overlap) is that species recognition based on drumming patterns

may be an epiphenomenon of factors unrelated to mating behaviour or sexual selection per se. Indeed, sexual selection may only mildly contribute to shaping drumming structure, and instead could involve multiple signalling modes including, e.g. colour or movement displays. This potentially indicates that the unique cues to species identity found in drumming signals constitute another example of magic traits[15–17], supporting the idea of an evolutionary tinkering of drumming structure.

Given the mechanical constraints imposed on the production of drumming signals[53,64], it is also possible that we underestimate the strength of the selection pressure to increase species-specific information. In this higher selection pressure scenario, woodpecker species would simply not be able to respond to such a high selection pressure, and thus remain constrained in their ability to evolve their signals. While this could support the relatively low drumming divergence and imperfect species-identity encoding found in woodpeckers' drumming, it may only be true for this particular clade and signal. The lack of a direct fossil record for drumming behaviour (as is the case for most behavioural traits) is of course a limit here. Should future research clearly identify strong anatomical correlates of drumming acoustic features, valuable insights could be added to the picture of drumming evolution we provide here (note, however, that this will in any case be particularly challenging given the scarcity of woodpecker fossils in general[50,65]).

In summary, our approach reveals that species-specific acoustic signatures may not always be driven solely by acoustic divergence and that selection for species recognition needs not be strong, as long as it is adapted to local ecological requirements. This absence of systematic signal divergence during a clade radiation may be widespread among animals. We suggest that, similar to the quantification of species-specific information applied here, information-theoretic approaches[42] should be used more widely when studying the evolution of communication systems[66]. This theoretical ground provides a rigorous framework to examine a set of communication signals taken together (e.g. species-specific calls) and how systematic errors in decoding such signals can be used to increase the information being communicated.

As highlighted in this study, the Mathematical Theory of Communication[42] allows formulating hypotheses that can then be tested experimentally through the combined use of natural and resynthesized signals. It also has the advantage of taking into account the entire communication chain, from the emitted signal to the information decoded by the receiver[66], and should serve the development of analytical frameworks in the context of multi-modal signalling[67]. Finally, information-theoretic approaches can benefit existing research avenues such as those exploring the concept of 'meaning' associated with biological information[68,69], including information rate encryption in human speech[70,71].

## Methods

**Acoustic data and analysis**. Audio data were collected from online sound archives (Xeno-Canto—https://www.xeno-canto.org—and Macaulay libraries—https://www.macaulaylibrary.org), creating a pool of over 2000 audio tracks. We assessed the sound quality of these audio tracks by listening and through visual inspection of sound spectrograms. To capture intra-specific variation, we limited audio extraction to one drum per audio track (which also avoided pseudo-replication) and only included species for which at least 3 high-quality drums could be extracted. We retained 736 high-quality drums suitable for further analyses. These drums were distributed among 92 species (out of the 217 recognized species of woodpeckers[50] and 22 genera, providing a representative sampling of the phylogenetic diversity found in this family (Fig.1b)). Background noise and other artifacts were reduced by wavelet continuous reconstruction (R 'WaveletComp' package[72]), following the methods and description outlined in previous work[73]. The full script is available on demand. Finally, 22 acoustic variables were extracted from these filtered sound samples using the R 'Seewave' package[74]. Given the pulsed-like nature of drumming, the chosen variables emphasized the temporal and amplitude-related (all normalized to the maximal amplitude within a given drumming signal) features of the sounds (Supplementary Table 1). These 22 variables were z-scored and then used in all subsequent analyses. Since these variables were partly correlated to varying degrees, we performed a principal component analysis (PCA) to reduce the number of descriptive variables quantifying drumming acoustic structure. This dimensionality reduction was useful for visualization and necessary for regularization (i.e. to prevent overfitting) in many of our analyses. This resulted in six principal components (PCs) with eigenvalues >1 which together explained 75% of the variance (Supplementary Table 11).

We used these variables to evaluate the similarity between species-specific drums, by performing a hierarchical cluster analysis (HCA)[75,76] based on Euclidean distances and the 'Ward.D2' method ('NbClust' R package[77]). NbClust provides a clustering output resulting from the use of multiple indices (in the case of our analysis, 26 indices were used). The best number of clusters is chosen according to the majority rule, i.e. it is the one supported by the highest number of indices used. This entails creating a vector of acoustic features (22 raw acoustic measures or 22 PCs) for each of the 92 species in our dataset and calculating the Euclidean distance between these vectors to evaluate how close acoustically species were. Note that one can still use Euclidian distances in a non-orthonormal space to calculate the 'distance' between signals. The result is a distance metric that might give more weights to measures that co-vary. This could theoretically affect the clustering results. However, when we performed the same analysis using the 22 PCs, we obtained the same grouping (6 clusters) with very minor differences in species grouping and distances between clusters as shown in the relative length of branches (Supplementary Fig. 13). The output of this HCA established an optimal classification of woodpeckers' drums into 6 main drumming types (Fig. 2a), described as follows:

- Acceleration (AC): Beak strikes decrease in amplitude as they are produced within successively shorter time intervals.
- Regular sequence (RS): Beak strikes are produced in bouts, each comprising a relatively fixed (stereotyped) number of strikes.
- Irregular sequence (IS): Beak strikes are produced in bouts, each comprising a variable number of strikes (as opposed to RS).
- Steady fast (SF): Beak strikes are produced with constant time intervals and at a similar amplitude, with a high pulse rate (on average >20 strikes/s).
- Steady slow (SS): Beak strikes are produced with constant time intervals and at a similar amplitude, with a low pulse rate (on average <20 strikes/s.

In order to make an initial assessment of drum types' discriminability (which, if strong, can suggest a potential to encode species-specific information), we visualized the acoustic space occupied by the drumming of different species by plotting their spatial distribution in the 3d acoustic subspace spanned by the first 3 PCs; Supplementary Fig. 1a). We proceeded similarly using the linear discriminants (LDs) resulting from the DFA carried out to calculate species-specific drumming information content (Supplementary Table 12; Supplementary Fig. 1b; see next section for details on information calculation). This approach was conducted in addition to using the PCs to further validate our assessment of species information encoding in woodpeckers' drumming.

**Life-history data**. We used Gorman's specific description[44] to distinguish between species that produce drumming behaviour and those that do not (Fig. 1b). We attributed a 'drummer', a 'non-drummer' or an 'occasional drummer' status when this was clearly stated, and an 'unknown' status when the case seemed ambiguous (e.g. when conditional tense was used or when no clear report could be documented). We could thus define a drumming status for each of the 209 species used in reconstructing the ancestral state of this trait.

Similarly, distribution areas in square kilometres were obtained using the same source[44] in combination with the website 'https://www.daftlogic.com/projects-google-maps-area-calculator-tool.htm' for each of the 92 species included in our acoustic and phylogenetic analyses. Based on these distribution areas, a species pair was defined as sympatric as soon as an overlap was found between the pair's distributions areas, even if this was only at the range edge. We are aware that higher encounter rates (and thus potentially larger overlapping areas) are more likely to trigger a significant selection pressure for signal divergence between two species. Yet, this approach allows us to be conservative in the criteria used to defined sympatry (e.g. the difficulty of estimating an overlap percentage is much more likely to induce biases), and was supported by matching sympatry levels between our definition and the composition of the communities used in this study. Therefore, the 'sympatry level' used in our PGLS regressions corresponds to the number of species sympatric to a given species (e.g. *Veniliornins callonotus*' (*V.cal*) sympatry level is '4', i.e. 4 other species in our remaining sample (n = 91 other species) have distribution areas overlapping with that of *V.cal* (see Supplementary Data 2).

To assess whether morphological features determine drumming acoustic structure, we collected anatomical measurements on specimens from the Muséum national d'Histoire naturelle (MNHN, Paris, France) and the Natural History Museum (NHM, Tring, UK). We measured beak length, width and height (standardized, as measured at the most posterior point of the beak opening), the wing chord (from the most prominent point of the wrist joint to the most prominent point of the longest primary feather), and tarsus length on its inferior side. After initial inspection of inter-variable correlation, we retained beak length as the single beak measurement, and wing length as the single body size proxy measure[78], both to be used in further phylogenetic analyses (see section on PGLS). Wing chord was measured with an Ecotone ornithological ruler, while beak and tarsus measurements were collected using digital calipers (±0.02 mm accuracy for <10 mm measurements and ±0.03 mm accuracy for >10 mm measurements).

We calculated the 'beak length/wing length' ratio as a proxy for mechanical constraints on drumming. Drumming can indeed be physically considered as an 'oscillating spring', whose motion can be influenced both by beak length and body size[53]. To standardize our approach and match it to what was done for extraction of drumming acoustic features, we collected measurements from 3 specimens per species and computed median specific value for later analyses. Note that only one specimen was available for *Celeus spectabilis* (the type specimen) and *Picumnus nebulosus*.

Body size and body mass data were collected using literature data[44,79] and the following websites/archives for species with missing weight data: National Geographic, May 2015 (*Dendrocopos noguchii*); http://portal.vertnet.org/search?q=Campephilus+pollens (*Campephilus pollens*). Since we already had a measured proxy for body size with wing chord, we retained only body mass for later PGLS analyses.

**Calculation of information**. We quantified the species-specific information encoded within each specific drum. To this end, we used two classification algorithms to evaluate the actual discriminative power of drumming signals (and not only that of the classification algorithms), namely a Random Forest classification (RFC) and a DFA. Although the Random Forest algorithm can capture arbitrary groupings of acoustic features, it behaved with lower efficiency in cross-validation (Supplementary Fig. 14). We therefore chose the DFA to generate the confusion matrix of the posterior probability of each drum in our sample (n = 736) as belonging to one of the 92 species being studied. These posterior probabilities were generated in a leave-one-out cross-validation procedure (i.e. 736 different linear discriminant classifiers were trained based on 735 calls to classify the one call that was left out). To prevent overfitting, the DFA was based on the 6 PCA-scores and to ensure equal weighing of each species the DFA was trained with a uniform prior. This resulted in a confusion matrix (Fig. 2b) with 16.5% of correct classification (average of the diagonal), which was significantly higher than expected by chance with 1.09% (pDFA: p < 0.001; Supplementary Fig. 3). From this matrix, we calculated the local mutual information value (measured in bits) based on Shannon's Entropy[42] and following the Eq. (1):

$$MI_L(X_A) = \sum_M \left[ p(X_M|X_A) * \log_2 \frac{p(X_M|X_A)}{p(X_M)} \right] \qquad (1)$$

where $MI_L$ is the local mutual information for a given species ($X_A$); $p(X_M|X_A)$ is the conditional probability of classifying a drum as belonging to the species $X_M$ (M for the model, here the DFA) given that the actual species is $X_A$; and $p(X_M)$ is the unconditional probability distribution of predicted species. $MI_L$ quantifies the discriminability of one particular species, $X_A$, by taking into account not only the probability of correct classification but also the distribution of classifications both correct and incorrect for that species, $p(X_M|X_A)$, in comparison to that obtained

for the entire dataset, $p(X_M)$. The presence of systematic errors can provide additional information that is taken into account in information theory.

The overall mutual information, $MI$, is then given by the average $MI_L$ over species, following Eq. (2):

$$MI = \sum_A [p(X_A)*MI_L(X_A)] = \frac{1}{n_S}\sum_A MI_L \qquad (2)$$

where $n_S$ is the number of species. $MI$ can also be related to an average probability of correct detection, $p_c$ assuming equal probability of misclassification across all species $X_M \neq X_A$ (i.e. for scenarios where there is not systematic errors) by inverting the relationship shown in Eq. (3):

$$MI = \log_2 n_s - (1 - p_c)*\log_2(n_s - 1) + p_c*\log_2 p_c + (1 - p_c)*\log_2(1 - p_c) \qquad (3)$$

We also used Eq. (3) to calculate information through evolutionary time (evolution-through-time plots) for hypothetical scenarios (see below 'Analytical simulations of selection for information' and Fig. 3c, d) assuming different time courses for $p_c$.

$MI_L$ and $MI$ can be normalized by their 'ceiling value', namely the maximum amount of species-specific mutual information potentially encoded while discriminating $n_S$ species: $MI_{ceil} = \log_2 n_s$. Here, $MI_{ceil} = \log_2(92) = 6.52$ bits. Ceiling information is reached when the percent of correct classification is 100% for all species. Normalized $MI$ (both overall and local) values range between 0 and 100%. Comparing the normalized $MI_L$ values across drumming types showed significant differences (Supplementary Fig. 2b), but these could be the result of the unequal number of species within each drumming type. To control for unequal sample size for that analysis, we also calculated the $MI_L$ values selecting randomly 5 species per drumming type (corresponding to the maximum number of species available for the RS and IS drumming types) and iterated this computation 100 times. Comparison of the mean (over $n = 100$ iterations) normalized $MI_L$ across drumming types (each comprising $n = 5$ species) showed similar results to those found using the full number of species available, with IS significantly encoding more species-specific information than the other drumming types, followed by RS-AC, and then by DK, SS and SF (Fig. 2c).

We also estimated a CI, also obtained from the DFA output and defined by Eq. (4). CI ranges from −100 (minimum classification: for A and B a given pair of species, A is never correctly classified into A and always misclassified into B, and B is never correctly classified into B and always misclassified into A) to 100 (maximum classification: A is always correctly classified into A and never misclassified into B, and B is always correctly classified into B and never misclassified into A).

$$CI(A, B) = \frac{[p(X_A|X_A) - p(X_B|X_A)] + [p(X_B|X_B) - p(X_A|X_B)]}{2} \times 100 \qquad (4)$$

CI is preferred over normalized $MI_L$ for investigating the effect of sympatry on signal information because it allows considering species pairwise discrimination that can be directly compared to pairwise acoustic or phylogenetic distances.

## Evolutionary analyses

*Phylogenetic generalized least squares.* As morphological and ecological factors can play a direct or indirect role in the evolutionary changes in the acoustic structure, we used phylogenetic generalized least squares (PGLS) to examine the current relationships between life-history variables and drumming's acoustic structure and amount of information. PGLS allow the quantification of these relationships after accounting for effects that could simply be the result of phylogenetic closeness[80]. Since Miles et al.[54] found body size to influence drumming speed and sexual dimorphism to influence drumming length, we gave particular attention to the effect of physical traits (wing length, the ratio of beak length to wing length, and body mass) on drumming structure as well as the effect of geographical distribution traits (sympatry level, size of distribution area) on drumming information (see 'Methods, Life-history data' section above for details on life-history variables). We used PC1-PC6 (the components of the PCA carried out on drumming acoustic parameters) as proxies for drumming acoustic structure and the normalized mutual information to quantify the information content about species identity.

PGLS regressions were fitted using restricted maximum likelihood (REML). Model comparison was based on inspection of the Akaike Information Criterion corrected for sample size (AICc), using the null model's AICc as reference and stepwise forward selection. The improvement of a model was deemed significant only for a decrease in AICc >2 (from the AICc of the null model to the AICc of the fitted model). Model summaries can be found in Supplementary Tables 3, 5 and 6). For variable standardization, prior to running any PGLS model, all life-history variables were z-scored.

For models showing an improvement compared to the null model (i.e. ΔAICc >2; see models Supplementary Tables 3–6), a likelihood ratio test (LRT) was conducted to test for the specific effect of predictor variables (Supplementary Table 4). Because both models (null and fitted) differ in their fixed effects, model comparison was performed on models fit by maximum likelihood (ML) with the phylogenetic correlation structure (Pagel's λ) fixed to the estimates obtained from initial fit by REML. The statistics reported for model comparison are likelihood ratios.

PGLS models testing for a relationship between life-history variables and drumming structure included either of PC1 to PC6 as the dependent variable to

investigate whether differences exist between these proxies for acoustic structure. Similarly, LDs were used to verify our results with these different loading combinations of drumming acoustic variables. No significant correlations were found between life-history traits and acoustic structure using LDs instead of PCs (Supplementary Table 6), indicating that the combination of structural variation captured by the LDs differed from that of the PCs, while not leading to fundamentally different conclusions. Similarly, no significant correlations were found between life-history traits and information content (no decrease in AICc >2; Supplementary Table 3), overall emphasizing that none of the variables investigated here (and which could have potentially affected drumming structure) seemed to have influenced species-specific information, or at least not directly.

*Ancestral states reconstructions.* We carried out two types of ancestral state reconstructions: discrete reconstruction of drumming status in Fig. 1b and of drumming types in Fig. 3a (using 'ace' from the R 'ape' package[81]), or continuous character reconstruction of drumming acoustic structure based on Brownian motion model (using 'fastanc' from the R 'phytools' package[82] in Supplementary Figs. 4 and 5) and using relaxed Brownian motion model (using 'rjmcmc.bm' from the R 'geiger'[83] package in Fig. 4a and Supplementary Figs. 8 and 9).

While evaluating the likelihood that drumming was already present at an early stage of woodpecker's phylogeny, we tried to represent the most complete tree of the family, based on very recent molecular data[50]. Note that strictly speaking, we evaluate the state at the root but at the next internal node, i.e. at the node including Picumninae and Picinae (the largest pie-chart in our Fig. 1b), as Wrynecks do no drum, and neither do honeyguides or barbets. To include species with an unknown drumming status in this discrete reconstruction, we attributed equal probability distribution between the 3 states (i.e. when the 'drummer state' of a species is unknown, the species is given, prior to ancestral state reconstruction, a 1/3 probability of belonging to each of the three categories 'drummer', 'occasional drummer' and 'non-drummer'). Stochastic mapping was performed under an MCMC model, sampling the rate matrix from its posterior distribution for Q ('Q = mcmc' in make.simmap function from the R 'phytools' package), with an equiprobable default prior at the root, and 200 simulations. Under a symmetrical model for the probability to change among the three states, scaled likelihood on woodpeckers' ancestral node indicated 56.4%, 38.3% and 5.3% probabilities of being a drummer, an occasional drummer and a non-drummer, respectively. This is in line with the fact that morphological adaptations for drilling (including reinforced rhamphotheca, frontal overhang and processus dorsalis pterygoidei) evolved in the ancestral lineage of Picumninae and Picinae[64].

To prevent overfitting, the discrete reconstructions for drumming types were estimated for six different rate models: equal rate model (ER), symmetric rate model (SYM), all rates difference model (ARD) and three sequential transition models based on the normalized $MI_L$ as measures of complexity as shown in Supplementary Fig. 7 (SF ↔ SS ↔ DK ↔ AC ↔ RS ↔ IS). These three models assumed (1) sequential and equal, (2) sequential and incremental and (3) sequential and reversed transition rates, respectively. The number of parameters for these 6 rate models were 25, 1, 15, 1, 2 and 10. The final regularized likelihoods of each ancestral states were then obtained by model averaging using Akaike weights.

Calculation of information at different evolutionary steps was carried out as an extension of the drumming types reconstruction described above. From the discrete ancestral reconstruction procedure, probability distributions of drumming types were obtained for each node of the phylogenetic tree. We then obtained probability distributions at 20 fixed time intervals (dt = 1 myr) by linear interpolation. Using these probability distributions, we sampled drumming types proportionally from extant species descending the node closest to the time interval to estimate ancestral information values. This bootstrap procedure was repeated 30 times in order to obtain reliable estimates of mean and standard error. In this manner, we obtained information-through-time plots. These plots quantify a putative diversity of drumming signals in the clade at a particular point in time. They are similar in spirit to the disparity-through-time plots that have been used to measure specific morphological diversity in a clade through time using phylogenetic trees based on molecular data in combination with morphological measures in extant species[84].

Continuous ancestral character trait reconstruction of drumming acoustic structure was carried out using either the six PCs that explain variation among the 22 drumming acoustic variables, or the six LDs that explain the variation in discriminating potential among the same variables (see above, 'Acoustic data and analysis' and 'Calculation of information' sections; Supplementary Figs. 4 and 5). The results and conclusions were similar for all PC's and since the PC1 component has strong loading of multiple acoustic variables and the highest acoustic structure variance explained (Supplementary Table 11) it serves well as an illustrative example. The measure of phylogenetic signal on continuous traits (i.e. the historical contingency between species-specific drums that renders a trait non-randomly distributed along the phylogenetic tree) was made using Pagel's lamba (Supplementary Table 2).

Reconstructing information content from raw $MI_L$ values would not have been biologically relevant since information calculation is based on the number of species involved, a factor that changes as branches merge going backward along the phylogenetic tree. We thus reconstructed $MI_L$ based on the normalized $MI_L$ values to avoid this pitfall. We used a Bayesian model implemented in the R package 'Geiger'[83] (model 'rbm' in the function 'rjmcmc.bm') to estimate branch-specific

rates of trait evolution (i.e. changes in rates through time and across lineages). In this method, a reversible jump Markov Chain Monte Carlo (MCMC) sampling algorithm is used to detect shifts in rates of continuous traits evolution under a relaxed Brownian motion model[85]. The results of the model fit were summarized by the branch-specific average rate, estimated from the posterior samples. To obtain relative variations in posterior average rates, drumming structure (PC1–PC6) and $MI_L$ were standardized, i.e. these traits were divided by their standard deviation prior to running the 'rbm' models.

**Analytical simulations of selection for information.** In Fig. 3c, we compared that reconstructed evolution of information to what might be expected in different scenarios to further support those conclusions. More specifically, we estimated the ancestral $MI$ for two simulated scenarios using an analytical model that describes species-specific information based on the probability of correct detection and the number of species (see 'Calculation of information' section). In the 'No Diversifying Selection' scenario (dark brown), the probability of correct detection for the initial pair of species, $p_2$, is first estimated from the data using the approach described in the main text. It is then assumed that that additional species are randomly just as different/similar than these original species pair, yielding a probability of correct detection through time given by $p_c(t) = p_2^{n_s(t)-1}$, where $n_s(t)$ is the number of species at a given time. In the 'Strong Diversifying Selection' scenario (light brown), the probability of correct detection estimated at the first time point in our reconstruction ($-20$ M years ago, 3 species) is kept constant, $p_c(t) = p_2$. In other words, the only species that survive would be species that can discriminate themselves from all other species equally than the currently existing species. The reconstructed (actual) scenario is found between these two extreme values, showing that the drumming types are clearly not random but were also not under high evolutionary pressure to increase species-specific information. New drumming types evolved and species within types used signals that were distinct enough to result in the maintenance of normalized $MI$.

In Supplementary Fig. 6, we showed that the non-normalized reconstructed $MI$ increased more rapidly when new drumming types appeared but that the normalized $MI$ was relatively constant, reflecting the fact that the appearance of novel drumming types could co-occur with rapid radiation and increase in species numbers.

**Playback experiments.** Initial preparation involved identifying and mapping the areas prone to high densities of great-spotted woodpeckers *Dendrocopos major*, the study species of this experimental phase, using GIS maps provided by the LPO (French Bird Protection Organization). *D. major* is commonly found in European forests, ranging from open coniferous to mature deciduous forests. Playback experiments were carried out on wild individuals around Saint-Etienne, France, during this species' breeding season (February–April 2017). All experiments were performed in accordance with relevant guidelines and regulations including French national guidelines, permits and regulations regarding animal care and experimental use (approval no. D42-218-0901, ENES lab agreement, Direction Départementale de la Protection des Populations, Préfecture du Rhône).

Two sets of experiments were conducted over the course of the breeding season, although we implemented the same general design which consisted in simulating a territorial intrusion. Playback stimuli tracks consisted of eight drums spread unevenly over about 60 s, aiming at representing the variation encountered in natural sequences (ref. [44] and personal observations). The first experiment (Exp. 1) aimed at investigating *D. major*'s response to conspecific vs. heterospecific drums. The other experiment (Exp. 2) aimed at investigating *D. major*'s response to drums from conspecifics vs. drums modified through acoustic manipulation (i.e. signal re-synthesis). *D. major* typically drums with an 'acceleration' pattern, which is mainly characterized by a shortening of the inter-strike time interval, a progressive decrease in strikes' amplitude, and a gradual change in spectral properties as strikes get faster and weaker.

In Exp. 1, we used a paired and randomized order design, presenting each focal individual with one *D. major* drum and one drum from one out of 4 different species: 2 of which have very different drumming patterns (*Picus canus* and *Dryobates minor*, both producing 'steady fast' drums), and 2 others which have similar (accelerating) drumming patterns (*Dendrocopos syriacus* and *Dendrocopos hyperythrus*). A potentially confounding factor (which is nevertheless in line with our phylogenetic analyses) lies in that the allopatric species producing a drum similar to that of *D. major* also happened to be closely related to our model species. We carried out 48 playback experiments (testing 24 individuals with one of 4 categories of paired signals).

In Exp. 2, we altered one of the 3 acoustic features described above or all of them together (thus having 4 categories of modified signals), using Praat sound analysis software[86]. The design was paired so that each focal individual was exposed to one conspecific drum and one modified drum, following a randomized presentation order. This led to 48 playback experiments (24 individuals, each tested with one of 4 categories of paired signals).

Within each of Exp. 1 and Exp. 2, tested individuals were all separated by at least 500 m, ensuring different identities since their territory sizes vary between 200 and 400 m[87,88]. Upon visual or aural detection of (an) active individual(s),

the experimenter set up an Anchor Megavox loudspeaker at about 1–1.5 m from ground level. The speaker was connected to an Edirol R-09 recorder (stimuli tracks were created and stored as WAV files, 44.1 kHz sampling frequency). Playbacks started at about 50 m from where the experimenter last saw or heard the focal individual. Following the work from Schuppe and colleagues[89], playback intensity was calibrated and kept at about 80 dB measured 1 m away from the speaker. Behavioural data collection started when the first drum of the stimuli track was broadcasted and lasted 10 min from that moment on. To document focal individuals' responses, notes were taken manually and continuously, while audio was recorded with a Sennheiser ME67 microphone mounted on a tripod and connected to a digital recorder (Zoom H4N, 44.1 kHz, 16 bit). If a response was elicited from multiple individuals in the area, only the one from a particular individual (ideally the one seen or heard before setting up the experiment) was monitored and used in further analyses. Six behavioural variables were reported, namely the number of screams, the number of drums, the approach (which was divided into three categories: 'within 25 m', '25–50 m' and 'further than 50 m') as well as the latencies to first scream and drum and the latency to closest approach. When no occurrence was observed for the first three behaviours, latencies were set by default to the maximum value, i.e. the duration of the full experiment (10 min = 600 s). To characterize *D. major*'s behaviour, a PCA was then performed on scaled/centred data, where we retained the first principal component ('Playback-PC1') as an indicator of the behavioural response's strength. A higher Playback-PC1 score indicates a stronger territorial response, i.e. more screams, a closer approach to the speaker and shorter latencies to these 2 behaviours. A second significant component resulted from this PCA, which represented the drumming's response (inversely related: a higher Playback-PC2 score indicates fewer drums and a longer latency to drum; see Supplementary Table 13). None of the pairwise comparisons were statistically significant for PC2, besides a stronger drumming response to drums resynthesized without temporal variation than to *D. major* drums (Supplementary Fig. 15). This can be explained by the fact that birds were tested during the breeding season. At this time, drumming behaviour is likely to occur more consistently and commonly across experiments, independently from the stimulus played back, while screams and approach do not occur unless threat of intrusion is clear. Therefore, we used Playback-PC1 to represent birds' behavioural response in our analysis (as it is in addition explaining much more variance in the behavioural data than Playback-PC2). Note that, as two playback sets were involved in this study, while we considered them independently in our statistical analysis, for standardization of the behavioural scale, we used the same polynomial equation. More specifically, the linear equation obtained from the loading scores of Exp. 1 was applied to the behavioural data of Exp. 2 for computation of Playback-PC2 scores.

Finally, distances were approximated during continuous note-taking and confirmed post-experimentally using a National Geographic 4*21 rangefinder (measurement accuracy: ±1 m up to 200 m). Sex was not documented as sometimes birds were not seen (but just heard drumming or calling back at our playback), which we nevertheless believe to be negligible since both sexes drum and are territorial in this monogamous species[44,90].

Statistical analyses tested for differential responses of focal birds to drums of their own species versus either another species or a modified resynthesized condition. A paired comparison design was used by means of LMMs and contrasts using R software ('lme4' and 'lsmeans' packages)[91,92]. LMMs included study day and time, order of presentation and focal bird identity as random factors, and tested for a fixed effect of the interaction between treatment and group of paired condition. Contrasts were then computed between treatments (i.e. conspecific versus non-conspecific drums) for each group (i.e. each paired testing condition, such as *D. major* versus *D. minor* for which $n = 6$ birds were exposed to paired playback presentations—see Fig. 5a, b). Before contrasts and using the 'lsmeans' function, a Tukey adjustment for multiple testing was used; two-sided statistics are reported.

**Reporting summary.** Further information on research design is available in the Nature Research Reporting Summary linked to this article.

## Data availability
The authors declare that all data supporting the findings of this study are available within the paper and its supplementary information files. Raw data files are included as Supplementary Data 1 & 2. Source data are provided with this paper, and are also accessible at the following address: https://github.com/garciamaxime/DrummingEvolution_SourceData_SourceCodes.

## Code availability
Custom codes used for the various analyses in this manuscript are available at https://github.com/garciamaxime/DrummingEvolution_SourceData_SourceCodes.

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

## Acknowledgements

We are grateful to M. Greenfield, M. Hauber, F. Pellegrino, D. Reby, the LPO Loire France, E. Vericel, A. Villain, Lynx Edicions, and Xeno-Canto and Macaulay libraries. This research was supported by a Fyssen Foundation post-doctoral grant (M.G.), the Institut universitaire de France (N.M.), the Labex CeLyA and the University of Lyon/Saint-Etienne.

## Author contributions

N.M. and F.S. designed the overall research concept; M.G., N.M., F.S and F.T. conceived and supervised the study; M.G. and T.M.C. performed the acoustic analysis with inputs from N.M., A.R. and F.S.; M.G. conducted the field work and collected morphological measurements and audio data; J.C., J.F., M.G. and F.T. performed the phylogenetic analysis; F.T. developed the information-theoretic mathematical calculations with inputs from M.G., N.M. and F.S.; M.G., N.M. and F.T. wrote the paper, with inputs from J.C., J.F., A.R., and F.S.

## Competing interests

The authors declare no competing interests.
