## [Peer Review File · Nature Communications]

Reviewers' Comments:

Reviewer #1:

Remarks to the Author:

Review of Garcia et al., woodpecker drumming macroevolution.

Key results: Please summarise what you consider to be the outstanding features of the work.

--I believe the authors are correct when they state that theirs is the first clade-level analysis of the patterns in both signal evolution and "information content" evolution. These linked traits are subject to processes that act at multiple temporal and spatial scales (namely phylogenetic constraints and past divergent selection coupled with potential assembly rules based on current local-level discrimination), necessitating a battery of tests and comparisons.

Validity: Does the manuscript have flaws which should prohibit its publication? If so, please provide details.

--I did not find any prohibitory flaws, but communication ecology is not my field. I do not know if the Shannon-index metric of mutual information based on a DF-based "confusion matrix" actually maps onto anything subject to selection, and the 16% correct classification rate based on 3 drums per species, and the fact the Random Forest approach to classifying species using the 22 traits did not work well both make me wonder a bit.

Originality and significance: If the conclusions are not original, please provide relevant references. On a more subjective note, do you feel that the results presented are of immediate interest to many people in your own discipline, and/or to people from several disciplines?

--I would say the study scores high on originality, and so researchers in behavioural ecology are likely to want to read this to see if can be applied to other systems; I do not know if the approach leads to significant insight.

Data & methodology: Please comment on the validity of the approach, quality of the data and quality of presentation.

--I have not reviewed the data, only the results. The input data (the recordings and the tree) seem trustworthy – given the spread of the calls on the tree, I do not think the tree internal structure is that important – it is clear much of the variation would be reconstructed to have been present early on under simple models – see next comment (A propos, perhaps the approach by Harmon Science 2003 might be useful here).

Appropriate use of statistics and treatment of uncertainties...

--The reconstructions are statistical tests: given all but the double knock call type arises multiple times, the transition rates among call types is clearly high: reconstructions will be imprecise (most internal nodes are very uncertain, as depicted). This makes the claims in the reconstructions in Figure 3 a-c a bit hard to evaluate fully.

Conclusions: Do you find that the conclusions and data interpretation are robust, valid and reliable?

--I cannot comment on the specifics of the information content, but I did ask a lot of questions as I read through it.

Suggested improvements: Please list additional experiments or data that could help strengthening the work in a revision.

--I have made a large number of suggestions on the manuscript itself, e.g. areas that need clarification and questions that bear consideration to improve accessibility and maybe potential impact.

Clarity and context: Is the abstract clear, accessible? Are abstract, introduction and conclusions appropriate?

--I have made some suggestions to the abstract: phrases like "the encoding of species information has remained stable" and "tinkering with a constrained acoustic space" were not clear to me. I believe what is meant is that "species calls do not seem to overlap more towards the present vs the past even though the evolutionary reconstruction suggests the range of acoustic space was fully explored early in the radiation." Or something like that...

Please indicate any particular part of the manuscript, data, or analyses that you feel is outside the scope of your expertise, or that you were unable to assess fully.

--I was not able to assess the creation of the mutual information index per species fully; indeed, this is one reason this review is so late. I hope someone from Ref. 23 or Prof. Ryan at UT is also reviewing this.

Reviewer #2:

Remarks to the Author:

This manuscript presents multiple lines of evidence to test the hypothesis that woodpecker drumming signals have been shaped by selection to increase species information/recognition as the clade was undergoing a radiation. The authors identified 6 different drumming types in the woodpecker radiation, and show that although these signals lead to more correct classifications than expected by chance, that overall the rate of miscalculations between the different drumming types is low, and that most errors occurred between closely related species (which presumably share the most similar drumming types). The authors next tested whether the emergence of novel signals throughout the evolution of the woodpecker clade led to an increase in information content allowing for increased ability to discriminate between species (using character state reconstructing on a phylogenetic tree). They found that an increase in information content was balanced by an increased number of species leading to no change in overall information content. Next, using playback experiments, they found that the focal species discriminated between conspecific signals and heterospecific signals of a different drumming type, but failed to discriminate between conspecific signals and heterospecific signals of the same drumming type, suggesting that signals of the same drumming type did not contain sufficient species-specific information to be distinguishable. Finally, by looking at the composition of communities, they found that species within communities contained diverse drumming types (presumably to facilitate species ID) but only found evidence for character displacement when closely related species occurred in sympatry. The low number of closely-related sympatric pairs led the authors to conclude that a mechanism other than acoustic competition (namely foraging-niche exclusion) is the likely driver of community composition.

Overall I think this is a really interesting paper with a comprehensive set of data and the overall approach is novel. While the paper is well written in general, I found that I struggled to understand how all of the conclusions from each section linked to the primary hypothesis as set out in the first paragraph. I suspect that any confusion on these points could be improved with some relatively straightforward revisions to the structure of the main text in the manuscript, with a more clear map for how the evidence provided supports (or not) the hypothesis that selection has shaped signal structure to increase the ability to discriminate between species during a clade radiation.

I found some of the conclusions not well supported by evidence supplied in this paper. For instance, in the conclusion on Lines 228-230 the authors write that "Indeed while genetic drift and morphological specialization linked to foraging behavior drive signal divergence, the mechanical constraints upon

signal production combined with the absence of learning appear to limit the available evolutionary landscape." However, there is little evidence provided to support this statement. Genetic drift is invoked as an explanation for why more phylogenetically divergent species have more acoustically divergent signals, but alternative explanations such as ecological selection are not discussed. Morphology was not found to predict signal structure (Lines 154-155). The observation that communities where signal divergence is high in ecological communities suggests that differences in acoustic structure are important for species identification. The authors seem to suggest that it is more likely that these communities formed because closely related species cannot co-occur due to foraging niche overlap rather than signal overlap, but the observation that character displacement was only observed where closely related species are sympatric seems to contradict this statement. Again perhaps I have missed something which might be cleared up with a better explanation of how the results here support the hypotheses proposed at the start of the manuscript.

Specific comments:

Line 64: "evolutionary constraints leading to signal divergence" – sounds odd as constraints limit rather than promote - do you mean constraints limiting signal divergence?

Line 79-80: "Drumming is an innate behavior...with little evidence of having been shaped by sexual selection except for its duration and cadence" – It is not immediately clear what other properties could meaningfully be altered in a drumming signal. It would help here to provide more detail on the characteristics of drumming signals – how do these typically vary within and between signals?

Lines 112-115: "the amount of species-specific information decreases with the number of species sharing the same drumming type, and most errors in classification occur between closely related species." – Is this still important if species are not sympatric? As later said, closely related species occur in different niches due to foraging overlap (although not quantified here) so there wouldn't be any selection pressure to diverge?

Lines 141-143: "drumming signals do not appear to have been under high evolutionary pressure to maximize information for species identity, nor they did randomly drift as this would have resulted in a decreased in information" – how can you distinguish this from the alternative that they are constrained in their ability to evolve their signals? So there may be high selection pressure, but can't respond to that pressure?

Lines 145-150: Here the authors state that "drumming types did not appear in an order which could have increased their amount of species identity information." This assumes that the current drumming style of a species has not changed in the millions of years since that species appeared – how likely is that? Is there evidence from other studies that woodpecker signals are relatively constant? What role might ecology have played here over the millions of years of evolution? Is there any evidence from previous studies that drums are shaped by the environment?

Line 155: I was confused by the use of the term "socio-environmental variables" as this simply referred to whether species occurred in sympatry or not, is that correct or did I miss something here?

Line 186: I would expect the focal bird to respond more strongly if the amplitude was overall louder in these trials as a higher amplitude would be similar to an intruder who is nearby.

Line 208-211: It is not surprising to me that character displacement occurred only where two closely related species are in sympatry as the divergence in signals would then be reinforced by the negative consequences of making a misclassification error.

Lines 214-217: “as a result of random drift, distantly related species produce more dissimilar drums and are significantly less misclassified than closely related species” – can you rule out ecological speciation here?

Line 343: how was amplitude of signals measured in this study? Amplitude is notoriously hard to measure from field recordings, although it can be done more easily if the comparison is within a given signal rather than between signals.

Lines 375-383: Given that sympatry is an important aspect of this study, I would expect more explanation of how sympatry was defined here. How much of the distribution needs to overlap before a pair is considered to be sympatric? Are there some species pairs where the overlap is only at the range edge of one or more of the two species, meaning that population sizes (and likelihood of co-occurrence) are both likely to be low? For there to be significant selection pressure for signal divergence between two species, there needs to be a relatively high likelihood that these two species will encounter each other at a significant rate.

Line 592: by “maximal” do the authors mean “closest”?

Reviewer #3:

Remarks to the Author:

Garcia et al. describe a broad-based study of drumming signals in 92 species of woodpecker, with a focus on species identity as it relates to mate choice and competitor recognition. They also test for selection on signal structural divergence and species information during clade radiation, which is obviously different than signal divergence as it relates to mating and competition. The data set includes the phylogeny of drumming behavior and drumming signal types, information content of the signal at several levels, a playback experiment to one species, and a finer analysis of signal distinguishability in sympatric species.

I should say up front that this is a terrific study about an interesting signal. The scope of the study is outstanding. Drumming signals are a good choice for the study of signal evolution because their dimensionality is low (22 measured variables notwithstanding) compared to bird song. This makes the entire system more easily characterized. It also makes the approach more robust.

I should point out that there is one thing about the paper that I find a bit misleading. The framework is mostly about species identity. For example, the authors show (line 105) that the information content in the signals generates only 16.5% correct species classification when the entire 92 species data set is analyzed. Most of the information presented in this part of the paper is useful. However, it is valuable in a phylogenetic context, not in a species-identification context. This is because species identity is irrelevant for allopatric populations. Not until line 193 do we get an analysis of ecologically relevant species identity cues of sympatric species. However, the context of species identity is used throughout the paper, not just in reference to signals in sympatric species. This also is the context for species radiations. In some respects, the mode of speciation (e.g. vicariance vs sympatry) will dictate how relevant information content is across a radiation, but this seems beyond the scope of this manuscript (whose scope is already a bit breathtaking).

My suggestion is for the authors to be more explicit about why mating/competitor signal identity is particularly relevant across all species included in the study (note: I personally couldn't answer this), or perhaps more usefully discuss the basis of the evolutionary patterns per se and then discuss how these patterns contribute to signal divergence at the community (sympatric) level. Indeed, the answer

is likely in their conclusion (line 218) about how low relatedness among sympatric species is caused by foraging competition. This would make unique species identity cues in drumming signals an epiphenomenon of factors unrelated to mating behavior per se.

I need to reiterate that this paper was fun to read. It is very well written (I found only 2 cryptic typos on lines 332 and 588). The scope of the study is terrific. However, I think that the framework of the entire study needs to be cleaned up. I do understand that the phylogenetic patterns set up the basis of the analysis of sympatric signal divergence. But a more in-depth discussion of the basis of the phylogenetic patterns at the scale of the entire phylogeny is needed, as is a cleaner discussion of the relevance of information content across a variety of scales (taxon down to community).

Dear Editor and reviewers,

We would like to thank the editor for providing us with the opportunity to revise and resubmit our manuscript to Nature Communications, and the reviewers for their extensive and valuable input on our study. Below, you will find a point-by-point response to the reviewer's comments, which we found highly constructive and whose added value greatly improved our manuscript. Our responses are highlighted in bold font.

Reviewer 1: pp. 1-17;

Reviewer 2: pp. 18-22;

Reviewer 3: pp. 22-23.

Reviewer #1 (Remarks to the Author): Review of Garcia et al., woodpecker drumming macroevolution.

Key results: Please summarise what you consider to be the outstanding features of the work. --I believe the authors are correct when they state that theirs is the first clade-level analysis of the patterns in both signal evolution and "information content" evolution. These linked traits are subject to processes that act at multiple temporal and spatial scales (namely phylogenetic constraints and past divergent selection coupled with potential assembly rules based on current local-level discrimination), necessitating a battery of tests and comparisons.

We are glad to read the positive stance of the reviewer towards our approach and the topic investigated.

Validity: Does the manuscript have flaws which should prohibit its publication? If so, please provide details. --I did not find any prohibitory flaws, but communication ecology is not my field. I do not know if the Shannon-index metric of mutual information based on a DF-based "confusion matrix" actually maps onto anything subject to selection, and the 16% correct classification rate based on 3 drums per species, and the fact the Random Forest approach to classifying species using the 22 traits did not work well both make me wonder a bit.

It is clear that efficient (or informative) communication signals have evolved and it has been postulated by many that they are subject to selection. Here we propose a rigorous way to test this hypothesis using principles of Information Theory. The reviewer is correct in pointing out that, by necessity, our estimation of information is based on certain assumptions (representation of specific sound features, discriminant analyses). We are not necessarily stating that all of these features are under selective pressure. Instead, given the limitations of our data set, we attempted to use a minimum number of assumptions to calculate information. We present this novel approach and show here that it provides interesting results on how signal information changes during evolution. It is a method that could be refined or constrained for other data sets.

We also understand the reviewer's concern about the classification rate. It is on a first assessment apparently quite low but this is exactly one of the issues that we address in

the manuscript. First, although the rate of correct classification is low (16.5%), the actual normalized mutual information (as a bit percentage of the ceiling value; 38%) obtained from the confusion matrix is much higher. This is possible because misclassifications are systematic among related species. Second, we show that in local communities these misclassifications are avoided. Third, the species identity recognition based on drumming is clearly not perfect (but present and we argue sufficient) as shown in our behavioral experiments. We have added a few sentences (Lines 124-134) at the beginning of the results sections that more explicitly explain how that 16% number of percent correct classification does not capture systematic misclassifications and how these need to be analyzed in an ecological context.

Originality and significance: If the conclusions are not original, please provide relevant references. On a more subjective note, do you feel that the results presented are of immediate interest to many people in your own discipline, and/or to people from several disciplines? --I would say the study scores high on originality, and so researchers in behavioural ecology are likely to want to read this to see if can be applied to other systems; I do not know if the approach leads to significant insight.

We are glad that the reviewer finds our work highly original. As regards its significance, we would like to point towards some of the key insights gained from this work and stated in our manuscript, namely:

- **the novelty of our methodological approach,**
- **the first demonstration that although information can be a driver in the evolution of communication signals, the changes do not necessarily result in higher information rates.**
- **the dual consideration of the producer's (acoustic code Figs 2 to 4) and receiver's perspectives (behavioral responses Fig 5) with regard to the evolution of a communication signal.**
- **the necessity of understanding local population ecology for interpreting evolutionary trends.**

Data & methodology: Please comment on the validity of the approach, quality of the data and quality of presentation. --I have not reviewed the data, only the results. The input data (the recordings and the tree) seem trustworthy – given the spread of the calls on the tree, I do not think the tree internal structure is that important – it is clear much of the variation would be reconstructed to have been present early on under simple models – see next comment (A propos, perhaps the approach by Harmon Science 2003 might be useful here).

We thank the reviewer for pointing out this very relevant paper. The reconstruction of the information through evolutionary time is indeed very similar to the disparity-through-time plots that were generated in Harmon (Science 2003). They both measure the variance in the clade of a phenotype (morphology or acoustic signal) based on extant species to reconstruct a phylogeny (molecular data) and the morphology from a very large fraction of all species in the clade. The method in Harmon is indeed (only slightly) more simple since it just samples from extant species found downstream of each node. Our calculation is similar but based on a weighted sampling that is derived by the tree structure (via the reconstructed probability of drumming type). We believe that this sampling is more accurate (see also our response to the next question for caveats) but it is clear that both methods would yield similar results. We now cite the relevant Harmon

paper in our methods given the similarity between our information-through-time and their disparity-through-time plots (see Lines 888-893).

Appropriate use of statistics and treatment of uncertainties... --The reconstructions are statistical tests: given all but the double knock call type arises multiple times, the transition rates among call types is clearly high: reconstructions will be imprecise (most internal nodes are very uncertain, as depicted). This makes the claims in the reconstructions in Figure 3 a-c a bit hard to evaluate fully.

As the reviewer stated, the reconstructions of ancestral drumming types are statistical in nature. We also agree that transition rates can be high along certain branches (as we quantified in Fig 4 for PC1 of acoustic features for example). The probability of drumming types shown in Fig3 are estimates and the robustness of these estimates depends on four factors: the accuracy of the reconstructed phylogeny (high), the clustering of acoustical features, the relationship between the two that would affect transition rates, the number of species examined along each sub-clade. The distance in the hierarchical clustering tree data shows that the clustering of acoustical features into 6 groups is well justified. In addition, the phylogenetic clustering of these drumming types, although far from perfect, is also relatively strong (Supplementary Fig 4). Thus, while we agree that the reconstructed probabilities at some of the nodes might have high uncertainty (due to uncertainty in acoustical clustering and a small correlation between phylogeny and acoustic structure for that branch), the data shows that this is not the most common case: once a strategy has been identified at an internal node, it is generally found with high likelihoods across most subsequent lineages. This is also true after a transition has occurred (i.e. transitions do occur, but once a transition to a strategy has occurred, most subsequent lineages share this strategy). Thus, we do believe that it is relevant to analyze the reconstructed ancestral states (as probabilities) and use them in our information-through-time analyses.

It would be possible to attempt to estimate the robustness of our probability estimates by various simulations (e.g. using the probabilities of acoustical clustering or by removing particular species) but such simulations would require additional assumptions and we believe that the reader will be able to assess the reliability of these estimates given that all the underlying data is clearly presented.

Conclusions: Do you find that the conclusions and data interpretation are robust, valid and reliable? --I cannot comment on the specifics of the information content, but I did ask a lot of questions as I read through it.

In two sections below ('Manuscript annotations Reviewer #1' and 'Supplementary Material Annotations Reviewer #1'), we have addressed in full all comments made by the reviewer on the manuscript.

Suggested improvements: Please list additional experiments or data that could help strengthening the work in a revision. --I have made a large number of suggestions on the manuscript itself, e.g. areas that need clarification and questions that bear consideration to improve accessibility and maybe potential impact.

As authors, we really appreciate all the thorough work done by the reviewer. We know that this took a significant effort and we are the beneficiaries. Thank you.

All suggestions have been transposed from the manuscript to this letter, and are listed and addressed in the sections below named “Manuscript annotations Reviewer #1” and “Supplementary Material Annotations Reviewer #1”.

Clarity and context: Is the abstract clear, accessible? Are abstract, introduction and conclusions appropriate? --I have made some suggestions to the abstract: phrases like "the encoding of species information has remained stable" and "tinkering with a constrained acoustic space" were not clear to me. I believe what is means is that "species calls do not seem to overlap more towards the present vs the past even though the evolutionary reconstruction suggests the range of acoustic space was fully explored early in the radiation." Or something like that...

We thank the reviewer for their suggestions and have strongly revised the abstract, while also making sure to comply with the Nature Communications guidelines. In addition, we have reworked our manuscript while keeping in mind to place an emphasis on clarity of the approach and interpretations (Introduction, concluding remarks of our results subsections, discussion – various points are detailed in the specific manuscript’s comments below).

Please indicate any particular part of the manuscript, data, or analyses that you feel is outside the scope of your expertise, or that you were unable to assess fully. --I was not able to assess the creation of the mutual information index per species fully; indeed, this is one reason this review is so late. I hope someone from Ref. 23 or Prof. Ryan at UT is also reviewing this.

We appreciate this candid concern although, given the very pointed and accurate requests from the reviewer on the information theoretic sections of the manuscript, we actually believe that the reviewer had a good understanding of the measures that we are using and when she/he did not, it was principally because they were not clearly defined in our paper. We have attempted to address all the methodological questions the reviewer brought up and believe that this version of the paper is much clearer.

Manuscript annotations Reviewer #1

Line 25: replace woodpecker by ‘the clade’

Done.

Line 25: Why “only” - this suggests an expectation.

We agree with the reviewer and have removed ‘only’.

Lines 26-27: ‘interchanged’ and ‘tinkering’: would not “interchange” suggest “macromutations” rather than tinkering. It is not clear what the expected rate of tempo of change of drumming is that would allow the results to be considered “tinkering.”

By tinkering, we do not mean to involve a quantitative estimate of the rate of change of drumming, but rather attempt to provide a qualitative characterization of the unstable and interchangeable nature of drumming strategies. A key aspect of our reasoning, and hence the use of tinkering here, is that such changes (transitions between drumming strategies) occurred on various occasions *despite* a strong phylogenetic signal on

drumming structure.

We understand the reviewer’s concern, and to get rid of the confusion highlighted in the reviewer’s comment, we have modified this sentence by adding contextual information: “Acoustic analyses and evolutionary reconstructions showed that six main drumming types could interchange despite strong phylogenetic contingencies, suggesting evolutionary tinkering of drumming structure within a constrained acoustic space”.

Line 43: The definition in brackets is key, but includes a double negative “uncertainty reduction” that impedes clarity.

We use ‘uncertainty reduction’ following how information is defined by the Mathematical Theory of information. However, we agree with the reviewer and have rephrased this section between brackets to improve clarity (it now reads “the extent to which signals reduce uncertainty in the context of inter-specific discrimination”, lines 47-48).

Line 50: delete ‘a’ before ‘little effect’

Done

Lines 68-69: this is the novel aspect of this work, though of course it is not known if this framework is what the animals actually use.

The mutual-information makes assumptions regarding the choice of parameters that we use to describe the signals. Beyond that, information measure simply quantifies the discriminability of these signals and clearly this is relevant for animals. Note that we do not imply that animals try to ‘measure’ the information content of conspecific/heterospecific signals. Rather, we mean that they use the species-specific information available in their respective signals, which we try to quantify through mutual information calculation derived from a combination of parameters that we selected to be as representative of the signals as possible. In our experimental manipulations we also tested the relevance of our assumptions by parametrizing the acoustic signals. The novel aspect of this work is that signal diversity (both in terms of structure and information) in the clade is quantified and examined as a potential evolutionary driver.

Line 77: ‘Exapted’: not sure why one would use this term here. Evolved would work fine.

As suggested, we replaced ‘exapted’ by ‘evolved’.

Line 77-78: Add “...under a symmetrical model for the probability to change among the three states”.

We added this piece of information, however in the ‘Ancestral states reconstructions’ section of the Methods, where we believe it is better suited (lines 866-867).

Lines 104-108: This seems very important: the way the drums are scored here for information suggests they are NOT for species-specific recognition. Or am I missing something? Likely. It is not clear where the chance level comes from.

We believe we have already addressed this point in our reply above. In summary:

- 1. The 16% should be compared to chance which is 1/number of species or 1.9%**
- 2. The percent correct classification is not a good summary statistic because it ignores systematic errors and this is why we use mutual information.**

3. If systematic errors can be avoided, then the species recognition is much higher. We show that this occurs at the ecological level of the communities.
4. Finally, species recognition based on drumming is not perfect.

We have substantially reworded that section of the results to make sure that the reader has the overall picture from the beginning of the article (Lines 118-135). We also realized that this distinction between phylogenetic and ecological perspectives had not been sufficiently stressed in the previous version of our manuscript and have thus made an effort to clarify this throughout the manuscript.

Line 111: “irregular sequence” as a name suggests a grab bag of drumming types that do not conform to regular curve-fitting (a la the Am. Nat. paper) - this would also imply more information in that type, no? Almost by definition?

We understand the reviewer’s concern, however we have kept this name as is, for several reasons:

- the curve-fitting described in Miles et al.’s paper is based on cadence and acceleration to describe what the authors define as rhythm. We would like to point out that our analysis includes many more acoustic variables than used by Miles et al., including curve fitting variables (found in our Supplementary Table 1: *interval_slope_lm2_ordre_coefa* and *Amp_slope_lm2_ordre_coefa*) but not limited to those. While we appreciate the work carried out by Miles et al., we therefore believe that our acoustic characterization is more exhaustive and reaches a finer level, given its 22 dimensions (see Supplementary Table 1).

- We provide a clear definition of ‘irregular sequence’, which, given its acoustic characterization, is in fact *not* a grab bag, but rather corresponds to a homogenous type of drumming structure, defined precisely and in opposition to ‘regular sequences’. Applying this definition is what actually makes ‘irregular sequences’ different from other drumming types and generates its high information. Yet and to accentuate this opposition to ‘regular sequences’, we have specified it in the according Supplementary results’ section and have edited our main text to avoid carrying such an impression (lines 111-114).

- Finally, we would like to add that, if it were a drumming type that comprises all structures that could not be classified elsewhere, this would entail a category with a wide spectrum of acoustic structures, likely leading to a higher rate of classification errors with other drumming types, and thus to lower average information.

Line 113-114: Is this unexpected, given that drums in the same type are more similar (by definition)?

This is indeed to be expected. We have modified this sentence to include this remark (Line 141).

Line 116: This is not well-explained in the SM

We are now providing detailed explanation about the confusion matrix (this has actually been moved to the main document – now Figure 2b – and its explanation is given in the legend).

Line 120: I think figure 1 b could be used to show inertia in drums - there is inertia in types at least, and types are defined by structure.

We thank the reviewer for this suggestion and have added the figure 1b as a reference to the phylogenetic inertia (now line 148).

Line 123: ‘information complexity’: I think this is a new term.

We have removed this term, which we realize was too vague. The sentence now reads ‘we then evaluated changes in the amount of species-specific information’, Lines 152-153.

Lines 125-131: Aren’t both these inevitable? The types are defined. based on PCA and DF analyses. Given any phylogenetic inertia, perhaps even a white noise model, many of these patterns are inevitable - new types as defined by the PCA and clustering = more overall complexity, and, as lineages accrue within any type, there is a steady increase in information (though I am a bit hazy as to what that means exactly “increase in information”).

An increase of information (discrimination of drumming signals across species) is clearly not inevitable although it is indeed to be expected as long as the between-species versus within-species variance of the drumming signal remains constant and the number of species (or signals to be encoded) increases. What is more interesting to examine is the slope of increase after taking into account the expected result just from the increase in number of species. We found that slope is very close to zero and our data suggests that any increase in information (for example as a result of the emergence of new drumming types) is balance by the need for that additional information as the number of signals (species) to discriminate increase. We have rewritten this paragraph to very explicitly state what is and is not inevitable (Lines 154-176).

Lines 131-13: This is a rich but unsubstantiated evolutionary model of how drumming evolved; “new signals” = “new types” as defined by the clustering/PCA? Not sure what is meant by “the need for” here. Most speciation is allopatric. Actually, I am confused by these two concepts - “increase in information” vs. “more informative signals”

We agree with the reviewer that this section was unclear and that some of our word usage was confusing. As stated in our previous answer, we have restructured this section, clarifying our reasoning and avoiding confusion with information-related terminology and statements like “the need for”. The section now spans lines 151-181.

Line 134: This seems like an interesting result if true, so the null model has to be considered carefully.

Yes, the normalized mutual information decreases in a null model where pair-wise discrimination is maintained as the number of species increases. We have moved the methods for these analytical models (Lines 917-939) in the main paper and have added a cartoon figure to graphically show the assumptions (Figure 3c).

Line 141: My colleague Howard Rundle has impressed on me that the use of “selective pressure” is somewhat misleading. Different populations are in different selective regimes that change constantly: there is no wind or pressure from selection - selection is the result of differential fitness, etc...

We understand and agree with the reviewer that the use of ‘selective pressure’ needs to be well-informed. We believe that this is a matter of terminology and definitions, for which however no consensus has been met to date. As a result, and capitalizing on the

fact that this teleonomy is widely used and understood in the scientific community at large, we have kept the use of this wording here and in other places across the manuscript, while trying to avoid it whenever possible.

Line 142: change ‘nor they did’ to ‘nor did they’

We thank the reviewer for spotting this typo and have changed it accordingly.

Line 144: “The species information” - not sure what this is, particularly given the 16.5% classification rate.

This section has been reworded to make these terms more specific and (as explained above) we have also more explicitly addressed our various quantifications of classification (percent correct, mutual information and misclassifications at the local population level).

Line 152: ‘Species information’. That term again.

Fixed to be more specific (‘species-specific information’).

Lines 156-159: So much of this is in the SM, which is not really a sound structure: it seems like the main message is not in the main text...

We agree with the reviewer and thank her/him for raising this point. We have now brought some of this supplementary material back into the main document, in particular the results on evolutionary rates (now Figure 4; Lines 187-202) and more detailed account of our PGLS models (Lines 203-225). We have also moved the methods for the analytical simulations in the main text (Lines 917-939). Given the variety of approaches that we have used in this study, in order to keep the main message as clear as possible and focus our manuscript on the parts that provided significant added value/information, we have left PGLS results showing mild to no effect of life history traits as supplementary material.

Line 205: “I wonder what the correct test is: the same number of random species, correct? The test comparing discrimination within community to the whole clade is not relevant...so 4c,d should not be the main result - SF 11b is the correct one to consider, and it suggests that the communities are indeed no different from random collections, discrimination wise, and closer in actual distance due to phylo constraints SF11a?”

We believe that the test comparing discrimination within communities to the whole clade does matter. It clearly shows that species distribute their drumming structure within communities using the entire range used by the Clade. It further shows that discrimination is quite different in small communities and in the entire group. As mentioned by the reviewer this is mostly because the communities are composed of fewer species and, thus, there is less “opportunity” for making a wrong classification. This interpretation is now more clearly discussed in lines 267-275, which describes the current Figs. 6c and 6d (previously 4c,d).

In addition, the reviewer is right in highlighting the importance of SF11 (community simulations), which has now been moved to the main text, as Fig 6e and 6f. Moreover, we have improved those simulations by ensuring that the number of species used in the model communities are equal across all conditions. More specifically, in panels e and f, a balanced selection of 5 and 6 species per community was applied. The choice for such

numbers is not random: 5 species per community allowed us to include RS and IS when creating communities with only similar drum types (these two drumming types only include 5 species and therefore limited their use). In addition, choosing 6 species allowed us to include one drum type per species when creating communities with only different drumming types. The distributions for these simulations are now shown as cumulative distributions with error bars obtained by bootstrapping (100 iterations).

Line 206: This seems an important point, no reference here?

We apologize for this oversight and have added a reference for this statement, line 293.

Lines 209-214: “I think the Miles et al. test of the effects of sympatry is stronger than this one - ie. comparing only closely related species and classifying the pairs as sympatric or allopatric. Comparing sympatric species pairs to all species pairs confounds relatedness (time and biogeography) with sympatry-driven divergence, does it not?

Indeed, see the next sentence, which says as much...”

While we understand the reviewer’s point, we respectfully disagree with the fact that our approach confounds relatedness with sympatry-driven divergence:

Miles & colleagues only focus on a certain context which is that of closely related species. The strong effect of sympatry that they found is therefore not contradicted by our results, which also show an effect of sympatry for closely related species. However, our approach also highlights the fact that this effect is not found as species are more distantly related. In theory, an effect of sympatry could also have been found for distantly related species if they were found in similar geographical areas. In this sense, our study provides a broader perspective than the one used by Miles et al, as we are looking into the effect of sympatry at the clade level. By considering such a large-scale view, we also consider cases of species distantly related but which could have the same drumming strategy, and are thus able to provide more general conclusions, namely that other factors than sympatry could be the main drivers in the evolution of woodpecker’s drumming at the clade level.

Lines 220-223: So this is unnecessary conjecture.

We agree with the reviewer and have removed this section.

Lines 228-229: What is the evidence for this bold statement? The null model results?

We agree with the reviewer. This statement has been removed from the restructured discussion (we nevertheless still discuss, using more careful suggestions, the effects of genetic drift, morphological constraints and potential foraging niche exclusion), and the issue raised by the reviewer no longer appears in the text.

Lines 230-231: Again, was there an explicit test of “drumming type” as an “attractor”?

The reviewer is right and we apologize for this unfounded interpretation. We have now restructured the discussion and the notion of attractor does not appear in the revised version.

Lines 239-241: “our information-grounded approach leads to a comprehensive understanding of the history and significance of signal divergence within communication networks” ...because...This is a conclusion that I do not see reflecting the discussion, or even much of the results. What did we learn that we would not have learned if this approach had not been

used?

I am still tripped up by the poor performance of the species-specific signal discrimination (16%). I feel this was not explored fully enough -

With regards to our conclusions, we have now made a major restructuring of the discussion section, where we took into account the points raised by all reviewers to strengthen our assertions.

In this letter, now we have also provided detailed explanation as to why the 16.5% classification rate is actually *not* poor (see comments above,), and have rephrased the corresponding sections in the paper to make this come across more clearly.

Lines 260: ‘species recognition during mating and territorial defence’: And a display function too, no? The Myles et al. paper explicitly tests for its function in male display.

Yes. We have broadened the concept by rephrasing this section (now less centered on species recognition), lines 455-456.

Figure 3b: I would use “strong diversifying selectIon” and “no diversifying selection” here: “Pressure” is hard to decipher (and also technically incorrect when referring to selection and response to selection.

We thank the reviewer for this suggestion and have modified figure 3b accordingly.

Figure 3c: Though I do not believe n=92 observations can support 12 inferred transition rates, this is still an interesting graphic.

We are glad to read that the reviewer found this plot interesting. We were also aware of the danger of overfitting. For regularization, six different rate models were used with various number of parameters corresponding to different assumption on the transitions probabilities (all identical, serial, symmetric, serial and symmetric, fully connected). The number of parameters in the 6 models were: 1,1,2,10,15, and 25. The final result were obtained through model averaging based on the six transitions matrices and their respective model’s Akaike weight (as indicated in the methods, Lines 873-880). This plot has now been moved to the supplementary material (Sup. Fig. 7) after reconsidering its relevance to the manuscript.

Figure 4: Are the sympatric and non-sympatric lines different within panels? Are those differences different between panels?

Differences between sympatric and non-sympatric lines are considered within each panel separately. In order to avoid any confusion, we have now clarified this in the figure’s legend (now Figure 7; Lines 659-665).

Lines 323-324: “which are more likely to be affected by sympatry” Not sure what that means “which are more likely to be affected by sympatry”

Combining our response to this comment and the previous one, we have now added clarifications in the figure’s legend, lines 659-665.

Lines 346-349: “This is but one of very many possible clustering approaches. Ward.D2 seems reasonable, and the resulting phylogram does seem to show discrete clusters, but “majority rule” is not one of the indices named as such in NbClust; please expand.

The PCA plots of the 22 z-scored can be very informative.”

- The ‘majority rule’ given in the NbClust package relies on the fact that multiple

indices are used to evaluate the output of the clustering approach. The chosen output is the one supported by the highest number of indices. E.g. in our case, a clustering output of 6 clusters was supported by the highest number of indices and is therefore the number we chose. As suggested by the reviewer, we have now expanded on this in the manuscript, lines 692-695.

- We agree on the fact that PCA plots can be very informative, and they have indeed served our exploration of the data. However, we believe that this information is already in the current version of the manuscript in various forms. The loading scores are indicated in the Supplementary Table 11 and we also generated 3D plots showing the positions of each species (color coded by drumming style) in Supplementary Fig 1.

Lines 349-354: “This procedure (euclidean distance of the 22-dimensional vector) only makes sense if the 22 variables are themselves orthogonal, no? I might have done the PCA first to see where species sit in a fictitious acoustic space, and then take euclidean distances from that (which are good distances) to see how many clusters form. Should still be 6.”

The reviewer is correct that to obtain a Euclidian distance metric which is invariant to rotation requires orthonormal axes. Also our 22 acoustic features are not uncorrelated (see correlation matrix belo). However, one can still use Euclidian distances in a non-orthonormal space to calculate the “distance” between signals. The result is a distance metric that might give more weights to measures that co-vary (if positive and negative correlations don’t cancel out). This could affect the clustering results. However, when we performed the same analysis with PCA, we obtained the same grouping (6 clusters) as shown in the figure below with some very small differences in distances between clusters as shown in the relative length of branches (now also the supplementary figure 13).

Why not use PC's then? The use of the right metric for acoustical distances is currently

a debated topic in the analysis of vocalizations. Although PC or factor analysis could yield uncorrelated axis, they might not span the acoustical space occupied by a particular ensemble of vocalizations (e.g. Wadewitz et al., 2015). In other words, this subspace might not be well characterized by a multivariate normal. More recently researchers have suggested the use of non-linear embeddings using techniques such as UMAP (Sainburg et al. 2020). These embeddings are based on what is referred to a geodesic distances instead of Euclidian distances. We have also examined our data with such an embedding and found that the clustering obtained through NbClust and the 22 acoustical features also clearly segregates all drumming type groups in such embedding (thus, two completely different unsupervised algorithms yield congruent results, see Figure below). This grouping is therefore fairly robust to our choice of distances. For this paper, we have decided to just present the hierarchical tree obtained with the 22 correlated acoustical features in the main paper. It requires the least amount of data processing and yields groupings at multiple levels that are easy to interpret (for example double knock is similarly distant to steady fast and steady slow).

UMAP projection of the woodpecker drumming data

We do however describe some of this reasoning in our methods (lines 698-703) and have added an additional plot in supplemental sections to show the dendrogram obtained by Euclidian distances on PCs (Sup. Fig. 13).

References cited in this response:

- Wadewitz, P. et al. (2015) Characterizing vocal repertoires - hard vs. soft classification approaches. PLoS ONE 10 (4), e0125785.
- Sainburg et al (2020) BioRxiv doi: <https://doi.org/10.1101/870311> (soon in press)

Lines 357-360. I am not sure what this means: this is just the first 3 PCA axes from above, so does not speak to “species-specific information” or at least not specifically. The LDs from the DF on the information content is something else.

We agree with the reviewer and apologize for creating a shortcut here: this representation’s key point is to allow visualizing the discriminability among drumming types, which can then indirectly suggest a potential for species-specific information if discriminability is high. We have now modified this section accordingly, lines 705-707.

Line 421: ‘confusion matrix (Supplementary Fig. 2)’: This is the key input data for the rest of the study: please explain what the entries are: e.g. the posterior probability from the leave-one-out DFA analysis that a species (x) is classified as species (y), with column sums =1 (or perhaps vice-versa?)

We have now clarified these issues in our Fig. 2b’s legend (because the classification matrix has now been moved to the main document as Fig. 2b). The confusion matrix does indeed show conditional probabilities obtained from the leave-one-out CV. The rows are the actual species and the columns the prediction obtained from the DFA. Each row sums to 1. This is now well described in Fig. 2b’s legend.

Line 423: ‘1.09%’: best to say “1/92” here. Given that species differ, and a DFA was used with a leave-one-out cross validation, is this really the correct comparison? It would be almost impossible to produce a dataset that would not do a lot better than “chance”, no?

Chance is the expected value in cross-validated classification with completely random assignment of groups. As a sanity check, one can perform the same analysis with

permuted species labels and the histogram of percent correct for 1000 permutations is shown below (the null hypothesis; it is now the supplementary Fig. 3). Clearly this permutation test shows that the result is highly significant ($p < 0.001$; dashed line is the “chance”, and the actual value of 16.5% is far off the chart). But the reviewer might be questioning the effect size in the sense that any vocal signal will have some degree of species signature? A comparative approach using vocal signals from another clade could be a useful metric. In our original version, the focus of percent correct for the entire genus (16.5%) gave the wrong impression that this effect size was too small to be important. We now hope that the revised version more explicitly focuses on information and that the local analysis has addressed this shortcoming. We also agree that the species signature is not maximal, however it is significant, used, and clearly sufficient. This is one of the central messages of our paper.

Line 426: FORMULA: This reads as if MI and I_L are not the same quantity. Please clarify here, because I_L is called “local mutual information”, not “multial information value”
 Also, indexes R and M are not defined. Nor is “real” in “realspecies”. Do you mean “focal species”?

The local information is an intermediate step in the calculation of the mutual information and we use it to estimate a measure of discrimination for each species. To be very explicit, we now use ‘local mutual information’ and ‘overall mutual information’ (Lines 777-791). In the original version R stood for the Real identity of the species and M for the identity predicted by the Model. We have changed to A (for Actual species) and M (for Model prediction).

Most generally, these equations are not presented in the standard way. For instance I_L(X_R) seems to be “local mutual information for species R,” but looks like the product of two indexed quantities (I and X).

We believe that this is the way functions are usually used. To make it clearer, we have

adjusted the description in the text, which now reads ' $MI_L(X_A)$... where MI_L is the local mutual information for a given species (X_A)'. We have also used '*' signs to indicate products without ambiguity.

I would also have put brackets around what is being summed across M.
Done.

Finally, can one put I_{L_XR} into words from the point of view of species X_R ? This is some measure of the probability that it would hear its own species' call but mistake it for some other species? If that is the case, why is the average of such values called "TOTAL" information, rather than something that denotes an average? I can see how one can get excited by calling something "information," but bringing it back to the behavioral ecology seems important. I note that ref. 23 does a great job of doing that, ie. discussing how and why selection might lead to discrimination differing among species as a function of the acoustic environment, etc. I feel this much bigger paper here does not do that as well.

The local mutual information is indeed an indicator of how well a particular species can be distinguished from other species while taking into account the structure of the "errors". We now explain this more clearly in the methods (Lines 777-788). It is also one of the key advantages of using information theoretic metrics instead of just percent of correct classification. The errors are not random and quantifying this non-randomness is indeed importation from an ecological perspective.

The label 'total information' was not appropriate. We now contrast local mutual information (for one species) to overall mutual information (for all species) and changed the notation.

We have reworded the methods describing the mutual information (lines 777-788) and further stressed the advantages of using information theoretic approaches in the main manuscript's discussion.

Lines 427-428: "marginal sum of classification percentages for each species": Please define - across the rows or columns of the confusion matrix? In words, it is the sum of the probabilities that a drum from any other species would be considered the drum of X_R ?

That section has been completely rewritten to more clearly define the conditional and unconditional probabilities that are used in the mutual information calculation. The new text is found lines 777-788). We have also modified the legend of Fig 2b (see also response to comment below) to describe what is shown in the confusion matrix on that figure and how it relates to the conditional probabilities defined in the Methods.

Lines 439-440: "namely the maximum amount of information available": This is a funny (to me) construction from an evolutionary point of view: one would rather speak of "the minimum amount of information required to discriminate n_s species" as $\log_2(n_s)$, rather than "information available while discriminating n_s species"

We understand the reviewer's comment. The nuance is indeed subtle, but the ceiling value does not indicate a minimum amount of information necessary for discrimination. Species can in theory be discriminated effectively, even if not perfectly, with lower information values. Reaching the ceiling information value would entail a *perfect* (in the sense of maximal) discrimination, and would be visualized by a single dark-blue diagonal line in our confusion matrix. However, species-specific information needs not be perfect to allow species-specific discrimination, as shown in our paper both

theoretically and experimentally. Therefore, this ceiling value does represent a ‘maximum amount of information’ potentially encoded. To avoid confusion with the notion of ‘availability’, we have rephrased this section, lines 798-802. We now clearly describe what a ceiling value would mean: “Ceiling information is reached when the percent of correct classification is 100% for all species”.

Am I correct in thinking that the distance from $\log_2(n_s)$ is an empirical measure of the propensity to misclassify a particular species? IF so, this would link SM figures 2 and 3.

Yes, this is correct. As misclassification increases, the distance from the ceiling value increases but also depends on the structure of the misclassification. The distance will be larger for more random misclassifications. It does indeed link figures 2 and 3.

Line 501: symmetrical, or is occasional between none and always?

We have now clarified this point, lines 866-867 (also in response to the reviewer’s comment “Lines 77-78”).

Lines 502-503: Technical point, this is the marginal reconstruction (the default), yes, not the joint reconstruction.

And, I believe the default setting will have an equiprobable prior root state, though one might want to check that - at equilibrium, the prior on a state might be its observed frequency, meaning the majority observation will almost always be the most likely.

If you want to test the hypothesis that drumming was ancestral, just compare the tree likelihood when you set root to non-drummer vs. when you set it to “drummer,” integrating over the states at the other nodes. I think Nosil Evolution 2005 lays this out.

The default prior is indeed equiprobable at the root. Note that strictly speaking, we do evaluate the state at the root but at the next internal node, i.e. at the node including Picumninae and Picinae (the largest pie-chart in our tree, as Wrynecks do no drum, neither do honeyguides or barbets). This is now stated in the methods (Lines 857-859).

We also had failed to specify that we were using a simulation of potential values of the transition matrix in the ancestral reconstruction based on its posterior distribution using the ‘make.simmap’ function from the R ‘phytools’ package. These simulations provide a posterior estimation of the probability of being drummer, non-drummer or occasional drummer starting with that uniform prior. We believe that this approach provides more information on the reliability of the results that likelihood differences. This is also now described in the methods (Lines 864-869).

Supplementary Material Annotations Reviewer #1

- Supplementary Fig. 1: “While these are attractive, I do not think they do a good job of illustrating how well the axes discriminate the types: I would prefer three panels for each, so we can be convinced that there are, in fact, 6 “types”. I. think this characterisation of types is one key component to the paper.”

While we understand the reviewer’s comment, we have used 3D plots as this combines 2D representations and allows presenting results both using the PCs and the LDs without an excess of descriptive illustrations for the readership. To highlight this, please find below the panels showing the combination of three 2D-plots that are equivalent to one of our 3D-plots.

We would like to raise the point that this illustration (Sup Fig 1) is for visualization purposes only. Given that 6 principal components have been retrieved from our analysis, to provide the best illustration of how well drumming types are separated in the acoustic space, one would need a 6-dimension representation (corresponding to the 6 PCs), which is practically not feasible. From this theoretical 6D illustration, removing dimensions can only lead to a decrease of the potential to visually discriminate drum types, hence the observed decrease of *visualized discriminability* when moving from a 3D (Sup Fig 1) to a 2D panel (below). We have therefore kept our 3D representation. Yet, we thank the referee for highlighting the importance of characterizing drum types, and would like to state as a brief reminder that, while Sup Fig 1 offers a visual illustration, we have quantified this discriminability through our cluster analysis (Fig 2a).

- Supplementary Fig. 2: “This is missing critical information: what do the colours represent, what do the dots actually represent? the Legend refers to posterior probabilities, but I see no probabilities anywhere on the graph, so I guess they are coded in the colour scheme.”

We thank the reviewer for spotting this and apologize for the lack of clarity in the legend. We have now seen to this and provided detailed explanation about the confusion matrix, the color code and the relationship with the probabilities described in the Figure’s legend (now Fig 2b) and in the methods section on information theory (Lines 775-788).

- Supplementary Fig. 6: “Note, we could only perform this simulation for up to 5 M years ago because beyond that point in time the number of species in our tree is greater than the number of extant species in our data set within a single drumming type”. I do not understand this or its significance.

We modified the legend it now reads:

“Note, we could only perform this simulation for up to 5 M years ago because the simulation involves sampling from extant species sharing the same drumming type. Beyond 5 Myrs, the number of species sharing a single drumming type at present is smaller than the number of ancestral species in the clade (as determined by the phylogenetic tree).”

Reviewer #2 (Remarks to the Author):

- This manuscript presents multiple lines of evidence to test the hypothesis that woodpecker drumming signals have been shaped by selection to increase species information/recognition as the clade was undergoing a radiation. The authors identified 6 different drumming types in the woodpecker radiation, and show that although these signals lead to more correct classifications than expected by chance, that overall the rate of miscalculations between the different drumming types is low, and that most errors occurred between closely related species (which presumably share the most similar drumming types). The authors next tested whether the emergence of novel signals throughout the evolution of the woodpecker clade led to an increase information content allowing for increased ability to discriminate between species (using character state reconstructing on a phylogenetic tree). They found that an increase in information content was balanced by an increased number of species leading to no change in overall information content. Next, using playback experiments, they found that the focal species discriminated between conspecific signals and heterospecific signals of a different drumming type, but failed to discriminate between conspecific signals and heterospecific signals of the same drumming type, suggesting that signals of the same drumming type did not contain sufficient species-specific information to be distinguishable. Finally, by looking at the composition of communities, they found that species within communities contained diverse drumming types (presumably to facilitate species id) but only found evidence for character displacement when closely related species occurred in sympatry. The low number of closely-related sympatric pairs led the authors to conclude that a mechanism other than acoustic competition (namely foraging-niche exclusion) is the likely driver of community composition.

Overall I think this is a really interesting paper with a comprehensive set of data and the overall approach is novel. While the paper is well written in general, I found that I struggled to understand how all of the conclusions from each section linked to the primary hypothesis as set out in the first paragraph. I suspect that any confusion on these points could be improved with some relatively straightforward revisions to the structure of the main text in the manuscript, with a more clear map for how the evidence provided supports (or not) the hypothesis that selection has shaped signal structure to increase the ability to discriminate between species during a clade radiation.

We would like to thank the reviewer for their positive evaluation of the manuscript and highlighting the novelty of our approach. In line with their comment, we have significantly modified our text to more clearly formulate our core hypothesis and to maintain the connection between results and conclusions more directly connected to that hypothesis. The abstract, end of introduction, concluding remarks within results subsections and discussion were all significantly restructured. We are convinced that the manuscript reads much better after this revision and thank the reviewer for their suggestion.

- I found some of the conclusions not well supported by evidence supplied in this paper. For instance, in the conclusion on Lines 228- 230 the authors write that “Indeed while genetic drift and morphological specialization linked to foraging behavior drive signal divergence, the mechanical constraints upon signal production combined with the absence of learning appear to limit the available evolutionary landscape.” However, there is little evidence provided to support this statement. Genetic drift is invoked as an explanation for why more phylogenetically divergent species have more acoustically divergent signals, but alternative

explanations such as ecological selection are not discussed. Morphology was not found to predict signal structure (Lines 154-155). The observation that communities where signal divergence is high in ecological communities suggests that differences in acoustic structure are important for species identification. The authors seem to suggest that it is more likely that these communities formed because closely related species cannot co-occur due to foraging niche overlap rather than signal overlap, but the observation that character displacement was only observed where closely related species are sympatric seems to contradict this statement. Again perhaps I have missed something which might be cleared up with a better explanation of how the results here support the hypotheses proposed at the start of the manuscript.

We understand the reviewer's concern, and this conclusion has also been challenged by another reviewer. We realize that we should have been more careful while phrasing our conclusions, and have now made additional effort now to clearly outline what belongs to fact-based conclusions as opposed to speculative interpretations. The discussion has therefore been strongly restructured and connections between results and hypothetical frameworks made clearer, in addition to also consider, when applicable, alternative explanations. Following the reviewer's comment, we have also made sure to include the effect of character displacement in our discussion.

Specific comments:

Line 64: “evolutionary constraints leading to signal divergence” – sounds odd as constraints limit rather than promote - do you mean constraints limiting signal divergence?

By evolutionary constraints, we actually mean evolutionary mechanisms. We have modified this sentence accordingly.

Line 79-80: Drumming is an innate behavior...with little evidence of having been shaped by sexual selection except for its duration and cadence” – It is not immediately clear what other properties could meaningfully be altered in a drumming signal. It would help here to provide more detail on the characteristics of drumming signals – how do these typically vary within and between signals?

Duration and cadence characterize the overall drumming patterns but do not account for subtler acoustic variation that could be as important in conveying various types of information to conspecifics (not only species identity but e.g. individual identity, individual fitness, hormonal state, body size, etc...). Such other features are e.g. amplitude related features, variation in cadence and amplitude (i.e. dynamic, rather than static acoustic traits), and/or sequencing in bouts. But more importantly, this sentence was poorly worded as we meant to convey that drumming signals could be modified by selection. Following the reviewers comment, we have restructured this section and reworded the sentence (Lines 85-87).

Lines 112-115: “the amount of species-specific information decreases with the number of species sharing the same drumming type, and most errors in classification occur between closely related species.” – Is this still important if species are not sympatric? As later said, closely related species occur in different niches due to foraging overlap (although not quantified here) so there wouldn't be any selection pressure to diverge?

We understand the reviewer's point and have made clear in our revised manuscript why considering species-identity information at the clade is also important. As developed in the manuscript, throughout the introduction and discussion: the phylogenetic-based approach allows us to consider the evolutionary patterns (such as random drift and the

strength of the phylogenetic signal on drumming structure) at the clade level. We are aware that encoding of species-identity information is not biologically relevant at the clade level since species have not evolved having to differentiate their drums from all other species on earth (note that we now also mention this in the text, lines 91-96). Yet, understanding the phylogenetic history of drumming structure and its associated information at the clade level remains important: the clade-wide confusion matrix allows us not only to examine the entire acoustic space that has been explored for species discrimination within the clade, but also to interpret the results found at the community level. The community-level analysis highlights the fine mechanisms through which species discrimination operates and builds up on the underlying phylogenetic signal found at the clade-level (now stated lines 330-336). Finally, we have rephrased more carefully the possibility (rather than presenting it as a fact) that foraging niche exclusion has an important role in these mechanisms (Lines 351-359).

Lines 141-143: “drumming signals do not appear to have been under high evolutionary pressure to maximize information for species identity, nor they did randomly drift as this would have resulted in a decreased in information” – how can you distinguish this from the alternative that they are constrained in their ability to evolve their signals? So there may be high selection pressure, but can’t respond to that pressure?

We agree with the reviewer and thank her/him for this insight. We have now included this point in our discussion, in connection with mechanical constraints applying on drumming production (Lines 365-367).

Lines 145-150: Here the authors state that “drumming types did not appear in an order which could have increased their amount of species identity information.” This assumes that the current drumming style of a species has not changed in the millions of years since that species appeared – how likely is that? Is there evidence from other studies that woodpecker signals are relatively constant? What role might ecology have played here over the millions of years of evolution? Is there any evidence from previous studies that drums are shaped by the environment?

The reviewer is right about this point and our reconstruction (from which our transition matrix is obtained) assumes that drumming style has not changed since the emergence of a given species. We do specify that ‘Drumming ... divergence has been relatively limited during woodpecker radiation’, citing the work from Miles & colleagues (2020). This being said, while of course subtle changes in drumming structures may have occurred (e.g. in parallel with changes in the habitat structure and thus of the trees available for a species to drum), the lack of knowledge about actual ancestral states is an implicit condition for any reconstruction of an acoustic trait, where no fossil record can be used to evaluate such changes. Note that in woodpeckers, a proxy for such fossil may be developed, only if strong determinants can be found between anatomical structures and drumming patterns (which is not yet the case). In any case, we now also stress this point in our discussion, Lines 369-373).

Miles, M., Schuppe, E. & Fuxjager, M. J. (2020) Selection for rhythm as a trigger for recursive evolution in the elaborate display system of woodpeckers. *Am. Nat.*, 195(5): 772-787.

Line 155: I was confused by the use of the term “socio-environmental variables” as this

simply referred to whether species occurred in sympatry or not, is that correct or did I miss something here?

We agree with the reviewer and have replaced it with a more suited term ('Geographical distribution variables', line 215), which refers both to sympatry and distribution area (another variable whose effect was considered in our PGLS – see Supplementary Material and Methods).

Line 186: I would expect the focal bird to respond more strongly if the amplitude was overall louder in these trials as a higher amplitude would be similar to an intruder who is nearby.

This is exactly our interpretation as well (the louder signal here representing in our opinion a 'superstimulus', *sensu* Tinbergen). We had left it out of the manuscript to avoid overloading the reader with details. However, following the reviewer's comment, we have now revised our choice and have added a word on this in the main text, lines 253-255.

Line 208-211: It is not surprising to me that character displacement occurred only where two closely related species are in sympatry as the divergence in signals would then be reinforced by the negative consequences of making a misclassification error.

We agree with the reviewer, and would like to stress that this is interesting in light of the lack of such an effect as phylogenetic distance increases. We thank the reviewer for this added interpretation, which we now include in our text, lines 303-307.

Lines 214-217: "as a result of random drift, distantly related species produce more dissimilar drums and are significantly less misclassified than closely related species" – can you rule out ecological speciation here?

We agree with the reviewer and apologize for this oversight. Regardless of the mechanism (the random drift that we suggested, or the ecological speciation suggested by the reviewer), the key message in this sentence lies in that the phylogenetic signal found on drumming structure supports our observations on the negligible effect of sympatry when phylogenetic distances are high between species. In the restructuring of our results, this section has been changed and we do not speculate on the mechanism in place to avoid misinterpretations (Lines 309-312).

Line 343: how was amplitude of signals measured in this study? Amplitude is notoriously hard to measure from field recordings, although it can be done more easily if the comparison is within a given signal rather than between signals.

Amplitude-related measurements are indeed often hard to measure. To circumvent this issue, all of our amplitude-related measurements are only made within (and not between) a given signal and are normalized to the maximal amplitude within this given signal to keep relative, instead of absolute measures. We have now added this methodological point in the manuscript, lines 681-682.

Lines 375-383: Given that sympatry is an important aspect of this study, I would expect more explanation of how sympatry was defined here. How much of the distribution needs to overlap before a pair is considered to be sympatric? Are there some species pairs where the overlap is only at the range edge of one or more of the two species, meaning that population sizes (and likelihood of co- occurrence) are both likely to be low? For there to be significant selection pressure for signal divergence between two species, there needs to be a relatively

high likelihood that these two species will encounter each other at a significant rate.

Following the reviewer's suggestion, we have now provided the requested details about how sympatry is defined here: A case of sympatry was defined as soon as an overlap was found between two species' distributions areas, even if this was only at the range edge. We are aware that higher encounter rates (and thus potentially larger overlapping areas) are more likely to trigger a significant selection pressure for signal divergence between two species. Yet, this approach allows us to be conservative in the criteria used to defined sympatry (e.g. the difficulty of estimating an overlap percentage is much more likely to induce biases), and was supported by matching sympatry levels between our definition and the composition of the communities used in this study. See lines 726-733.

Line 592: by "maximal" do the authors mean "closest"?

Yes. This is now fixed following the reviewer's suggestion.

Reviewer #3 (Remarks to the Author):

- Garcia et al. describe a broad-based study of drumming signals in 92 species of woodpecker, with a focus on species identity as it relates to mate choice and competitor recognition. They also test for selection on signal structural divergence and species information during clade radiation, which is obviously different than signal divergence as it relates to mating and competition. The data set includes the phylogeny of drumming behavior and drumming signal types, information content of the signal at several levels, a playback experiment to one species, and a finer analysis of signal distinguishability in sympatric species.

I should say up front that this is a terrific study about an interesting signal. The scope of the study is outstanding. Drumming signals are a good choice for the study of signal evolution because their dimensionality is low (22 measured variables notwithstanding) compared to bird song. This makes the entire system more easily characterized. It also makes the approach more robust.

We are delighted to read such positive feedback and thank the reviewer for their warm support to the topic and approach chosen in our work.

- I should point out that there is one thing about the paper that I find a bit misleading. The framework is mostly about species identity. For example, the authors show (line 105) that the information content in the signals generates only 16.5% correct species classification when the entire 92 species data set is analyzed. Most of the information presented in this part of the paper is useful. However, it is valuable in a phylogenetic context, not in a species-identification context. This is because species identity is irrelevant for allopatric populations. Not until line 193 do we get an analysis of ecologically relevant species identity cues of sympatric species. However, the context of species identity is used throughout the paper, not just in reference to signals in sympatric species. This also is the context for species radiations. In some respects, the mode of speciation (e.g. vicariance vs sympatry) will dictate how relevant information content is across a radiation, but this seems beyond the scope of this

manuscript (whose scope is already a bit breathtaking).

My suggestion is for the authors to be more explicit about why mating/competitor signal identity is particularly relevant across all species included in the study (note: I personally couldn't answer this), or perhaps more usefully discuss the basis of the evolutionary patterns per se and then discuss how these patterns contribute to signal divergence at the community (sympatric) level. Indeed, the answer is likely in their conclusion (line 218) about how low relatedness among sympatric species is caused by foraging competition. This would make unique species identity cues in drumming signals an epiphenomenon of factors unrelated to mating behavior per se.

We would like to thank the reviewer for this detailed comment as to how we could improve our manuscript. We agree with their suggestion and have put an effort into laying out more clearly the duality of our approach (i.e. phylogenetic vs ecological, lines 91-96) and how valuable it is (Lines 330-336). We have also included the reviewer's reasoning about how species identity cues could be an epiphenomenon of factors unrelated to mating behaviour per se and thank her/him for this valuable insight (lines 357-362).

- I need to reiterate that this paper was fun to read. It is very well written (I found only 2 cryptic typos on lines 332 and 588). The scope of the study is terrific. However, I think that the framework of the entire study needs to be cleaned up. I do understand that the phylogenetic patterns set up the basis of the analysis of sympatric signal divergence. But a more in-depth discussion of the basis of the phylogenetic patterns at the scale of the entire phylogeny is needed, as is a cleaner discussion of the relevance of information content across a variety of scales (taxon down to community).

Once again we would like to thank the reviewer for their warm support.

Typos line 332 and 588 have been fixed.

As before, we agree with the reviewer suggestion and, in line with the restructuring carried out in response to all referees, we have significantly modified the abstract, the end of our introduction, the conclusions of our results subsections and the discussion. In particular, we now emphasize the dual approach made at the clade and community levels (corresponding to an analysis of signal evolution at the phylogenetic and ecological levels, respectively).

Reviewers' Comments:

Reviewer #2:

Remarks to the Author:

I find this version of the manuscript much improved overall, and most of my queries have been thoroughly addressed. The introduction and discussion are particularly much improved and I agree with one of the other reviewer's comments that this paper is fun to read and it represents a very thorough analysis of the evolution of drumming signals. However, as a consequence of being a comprehensive study, there is a lot going on in this paper and it is really important to clearly describe predictions about what would be expected if drumming signals were under strong selection pressure to evolve to increase information content so that the reader can more easily judge whether the data supports those predictions. While this version of the manuscript does a much better job at summarizing key points and illustrating the connection between the different sections, I think a little more editing could help to make it easier to interpret the results.

I would start by suggesting that in each section (particularly the more complicated sections on information theory and evolutionary reconstructions) a clearer explanation of what the predictions would be for each set of analyses be included, before presenting the results (a sentence in each case would likely do). I highlight a few other places below where I think a little more careful editing could help to improve the clarity of the results.

Line 119: give a clearer explanation of "mutual information values" - I know this is explained more clearly in the supplementary methods, but a sentence here that summarizes how these values are calculated would help. How would these values look like if signals evolved for higher information content? What does MI = 38% tell us? Does 100% mean that the signals are so different that no mistakes are ever made?

Line 135 "[information for species identify in the drumming signal] is non-negligible and could provide reliable species discrimination as long as particular errors of classifications within related species could be avoided" - what does non-negligible mean here?

Lines 138-140: "To better understand which signal features provided information, we examined whether drumming types could have evolved as acoustic strategies supporting discrimination between species" - what analyses, and what would you expect to see if so? IS encodes more information - does this mean that there is more variability within this drumming type?

Lines 163-165: Why is this remarkable? I am not challenging that statement, I just think it would be clearer to state here what this result implies about the evolution of signals - which I think is that selection acted to maintain information (ie the ability to discriminate between signals) content of signals. It does come up later, but leaves the reader hanging for a bit.

Lines 187-188: I am not clear what "evolutionary tinkering" means in this context, or how this conclusion derives from the previous argument.

Lines 191-192: "The evolutionary rates of change in acoustic structure..."

Lines 246-247: Make it clear here in the text that the two species with the similar drumming technique were not sympatric with D minor.

Lines 263: It is still not clear to me why we would expect strong selective pressure at the clade level - I would expect this at the community level where species overlap, as has been shown here.

Line 296: overlapping not overlapping

Line 375: lack of fossil record for beak morphology rather than drumming behavior?

Reviewer #3:

Remarks to the Author:

I've gone over the manuscript and the reviewers' comments, focusing on reviewers 1 and 3. Unlike reviewer 1, I work on the ecology of communication and not on phylogenetics, so feedback on

reviewer 1's comments need to be considered in this light.

Reviewer 1 was concerned about the relevance of an information-theoretical framework. Shannon entropy has been used to study animal communication for a fairly long time and is used appropriately in this manuscript. The fact that the bit content of drumming signals is relatively low (resulting in a relatively low correct classification rate) is to be expected of a signal with a low dimensionality, certainly compared to signals such as bird song or even some bird calls.

Overall I think that the manuscript is cleaner and better. I generally agree with the responses to the reviewers. There is, however, one aspect of the manuscript that I still find problematic – the implication that drumming primarily functions for species recognition, although the logic is altered later in the manuscript. Instead, I don't think that there is any question that the primary function of drumming is as a sexually-selected signal, or more importantly part of a multimodal signal that includes color and movement patterns. Reviewer 1 brought this up (comment about line 260 – the response is supposed to be on lines 455-456, but this must be a typo because there is nothing in my version of the manuscript on these lines). Reviewer 2 brought this up about lines 79-80. The authors' response to this comment is on lines 85-87, where they note that that aspects of drumming duration and cadence have been shown to be sexually selected. However, the end of that sentence notes that they predict that signal structure should at least maintain species-specific information (i.e. selection on inter-specific interactions) during clade radiation and that this information will be sufficient to allow for discrimination of conspecifics from sympatric heterospecifics. I suggest that intra-specific selection will be stronger than inter-specific selection, particularly where niche differentiation limits the range of species that any given species will interact with. In addition, the authors address the possibility of sympatric speciation that might result from incipient changes in drumming structure (line 301). This assumes that the drumming pattern itself is the only signal that is sexually selected, which is simply not likely. Similarly, lines 333-335 state that "By reconstructing the evolutionary history of signal's information content in parallel to signal structure, our work adds and quantifies a functional perspective to evolutionary patterns, thereby offering novel insights into animal signal evolution." Again, the implication is that a strong function of drumming structure is inter-specific recognition. But then on line 350, they state that "...our results suggest that woodpeckers' drumming signal has not been selected for maximizing information for species identity in the signal itself but that it has nonetheless preserved the efficacy of its species' signature as the number of species in the clade increased." Then on line 363 the implication is that drumming patterns are unrelated to mating behavior and sexual selection.

In short: I still like this paper. I think that it could provide a very important example of the study of signal evolution in a system where the signal is simple but accessible to a broad experimental approach. However, from my perspective, the logic of the argument still needs to be cleaned up.

Minor points:

Line 35: make the sentence more explicit. "... species recognition in species assemblages that lack closely related species with concomitantly similar drum patterns".

Line 38: Would this work better? "... the effectiveness of information transfer relevant to inter-specific discrimination".

Line 298: "Through this process, we could expect phenotypic (in this case, drumming structure) differences to be as high for closely related species that share the same geographical area as for distantly related species or species that do not co-exist." "... or species that do not co-exist"? This needs to be clearer.

Line 370: Should "underestimate" be "overestimate"?

1034: "Tukey"?

REVIEWERS' COMMENTS:

Reviewer #2 (Remarks to the Author):

I find this version of the manuscript much improved overall, and most of my queries have been thoroughly addressed. The introduction and discussion are particularly much improved and I agree with one of the other reviewer's comments that this paper is fun to read and it represents a very thorough analysis of the evolution of drumming signals.

We are glad to read that our modifications met the reviewer's requirements, and also believe that the manuscript has strongly improved compared with its previous version.

However, as a consequence of being such a comprehensive study, there is a lot going on in this paper and it is really important to clearly describe predictions about what would be expected if drumming signals were under strong selection pressure to evolve to increase information content so that the reader can more easily judge whether the data supports those predictions. While this version of the manuscript does a much better job at summarizing key points and illustrating the connection between the different sections, I think a little more editing could help to make it easier to interpret the results.

I would start by suggesting that in each section (particularly the more complicated sections on information theory and evolutionary reconstructions) a clearer explanation of what the predictions would be for each set of analyses be included, before presenting the results (a sentence in each case would likely do). I highlight a few other places below where I think a little more careful editing could help to improve the clarity of the results.

We understand the reviewer's concern and had attempted to do so with clear predictions stated at the end of our introduction. To improve the clarity of our reasoning, we have followed the reviewer's suggestion and have edited the beginning of the suggested sections to provide specific predictions for each analysis:

- 'Information theoretic estimations': Lines 116-119. "Given the variation in acoustic structure of drumming patterns across the woodpecker family, we predicted that a bird species could be identified based on its drumming alone and used information theory to quantify the upper limit of one's performance in this species discrimination task", as well as, lines 158-163: "Since our hierarchical clustering analysis of acoustic structure revealed a finite number of distinct drumming strategies, we began to explore the relationship between acoustic structure and species signature by examining the contribution of drumming type to the MI for species discrimination. We postulated that novel (in the sense 'newly emerged') drumming types might evolve to increase the MI during clade radiation, and examined this hypothesis based on the 92 extant species in our data set"

- 'Evolutionary reconstructions': lines 181-184. "To do so, we produced evolutionary reconstructions of drumming types and of their associated information content (information-through-time plots; Fig 3a) along the woodpeckers' phylogenetic tree (see methods),

predicting that signal structure should have evolved to optimize species-specific information during the clade radiation”

We also edited the beginning of the ‘Information in ecological communities’ section: lines 295-297. “We then investigated further the hypothesis that woodpeckers living in sympatry evolve distinguishable drumming types, predicting character-displacement in cases where sympatric species shared a similar enough drumming structure”

Line 119: give a clearer explanation of “mutual information values” - I know this is explained more clearly in the supplementary methods, but a sentence here that summarizes how these values are calculated would help. How would these values look like if signals evolved for higher information content? What does $MI = 38\%$ tell us? Does 100% mean that the signals are so different that no mistakes are ever made?

We have added a few short sentences that explain in more intuitive terms what is measured in mutual information (lines 133 to 152). In combination, with the mathematical equations found in the methods, we believe that we have now a clear (and accessible) description of the information theoretic measures that we are using. 100% does indeed mean that the species can be perfectly identified based on their drumming signal.

Line 135 “[information for species identify in the drumming signal] is non-negligible and could provide reliable species discrimination as long as particular errors of classifications within related species could be avoided” – what does non-negligible mean here?

The term “non-negligible” is indeed vague and had made its way into our manuscript as a response to Reviewer #1 comments that the percent correct classification values were so low as to be negligible. We argue that correct classification does not fully capture species discrimination and that the actual value of information is much more useful for this purpose as it can take into account systematic misclassifications. In the reworded text, we have eliminated the vague ‘non-negligible’ qualification. Thank you for pointing it out.

Lines 138-140: “To better understand which signal features provided information, we examined whether drumming types could have evolved as acoustic strategies supporting discrimination between species” – what analyses, and what would you expect to see if so? IS encodes more information – does this mean that there is more variability within this drumming type?

We have now reworded this section to more clearly define the goals of this analysis and the explanation of the results. The analysis involves examining the contribution of drumming types to the mutual information calculated in the previous paragraph. IS encodes more information because it is a drumming type that is more different from other drumming types than the ‘average’ drumming type found across all species. The beginning of that paragraph now reads (lines 158-161): “Since our hierarchical clustering analysis of acoustic structure revealed a finite number of distinct drumming strategies, we began to explore the relationship between acoustic structure and species signature by examining the contribution of drumming type to the *MI* for species discrimination”

Lines 163-165: Why is this remarkable? I am not challenging that statement, I just think it would be clearer to state here what this result implies about the evolution of signals – which I think is that selection acted to maintain information (ie the ability to discriminate between signals) content of signals. It does come up later, but leaves the reader hanging for a bit.

We agree with the reviewer and have now modified the text accordingly. It now reads (lines 189-192): ‘Remarkably, this normalized species-specific information remained relatively constant during the woodpecker radiation (Fig 3b, brown curve), highlighting how selection pressures acted to maintain species discrimination even as the number of species increased along our phylogenetic reconstruction’.

Lines 187-188: I am not clear what “evolutionary tinkering” means in this context, or how this conclusion derives from the previous argument.

This relates to the fact that drumming seems to have alternated back-and-forth between types. We have rephrased this bit to clarify it (lines 214-216): ‘Instead, drumming types interchanged during woodpeckers’ radiation, such fluctuations reminding of an evolutionary tinkering of drumming structure within a constrained acoustic space’.

Lines 191-192: “The evolutionary rates of change in acoustic structure...”

The text has been modified according to the reviewer’s input. Thank you.

Lines 246-247: Make it clear here in the text that the two species with the similar drumming technique were not sympatric with D minor.

We would like to thank the reviewer for this suggestion. This is now done.

Lines 263: It is still not clear to me why we would expect strong selective pressure at the clade level – I would expect this at the community level where species overlap, as has been shown here.

We agree with the reviewer regarding the results found at the community level. In addition, as mentioned in the manuscript (hypothesis written in our introduction), the rationale of running analyses at the clade level lies in identifying a phylogenetic signal carried by woodpecker’s drumming, which helps understand evolutionary histories. This is also stated in the manuscript (lines 363-366): ‘Phylogenetic analyses allowed us to establish the broad-scale evolutionary patterns found within a clade radiation. This step is key for investigating how a signal’s acoustic space has been explored in a particular clade to represent species information, as well as for making predictions about the actual discrimination processes occurring at a biologically relevant scale’.

Indeed, one of the principal results found at the level of the community is that the acoustic diversity that can code species information is very similar to that which is found at the level of the clade; and, when closely related species are sympatric, their drumming has evolved to be more distinct preserving again the diversity found at the clade level. It is by performing both the analyses at the clade and community level that we can recognize these patterns and make predictions on their evolutionary histories.

Line 296: overlapping not overlapping
This has been fixed, thanks.

Line 375: lack of fossil record for beak morphology rather than drumming behavior?
Our point here is that there are no fossil records of the behavior itself. We have modified the text to make it more obvious (lines 404-405: 'The lack of a direct fossil record for drumming behavior (as is the case for most behavioural traits) ...')

Reviewer #3 (Remarks to the Author):

I've gone over the manuscript and the reviewers' comments, focusing on reviewers 1 and 3. Unlike reviewer 1, I work on the ecology of communication and not on phylogenetics, so feedback on reviewer 1's comments need to be considered in this light.

Reviewer 1 was concerned about the relevance of an information-theoretical framework. Shannon entropy has been used to study animal communication for a fairly long time and is used appropriately in this manuscript. The fact that the bit content of drumming signals is relatively low (resulting in a relatively low correct classification rate) is to be expected of a signal with a low dimensionality, certainly compared to signals such as bird song or even some bird calls.

Overall I think that the manuscript is cleaner and better. I generally agree with the responses to the reviewers. There is, however, one aspect of the manuscript that I still find problematic – the implication that drumming primarily functions for species recognition, although the logic is altered later in the manuscript. Instead, I don't think that there is any question that the primary function of drumming is as a sexually-selected signal, or more importantly part of a multimodal signal that includes color and movement patterns. Reviewer 1 brought this up (comment about line 260 – the response is supposed to be on lines 455-456, but this must be a typo because there is nothing in my version of the manuscript on these lines). Reviewer 2 brought this up about lines 79-80. The authors' response to this comment is on lines 85-87, where they note that that aspects of drumming duration and cadence have been shown to be sexually selected. However, the end of that sentence notes that they predict that signal structure should at least maintain species-specific information (i.e. selection on inter-specific interactions) during clade radiation and that this information will be sufficient to allow for discrimination of conspecifics from sympatric heterospecifics. I suggest that intra-specific selection will be stronger than inter-specific selection, particularly where niche differentiation limits the range of species that any given species will interact with. In addition, the authors address the possibility of sympatric speciation that might result from incipient changes in drumming structure (line 301). This assumes that the drumming pattern itself is the only signal that is sexually selected, which is simply not likely. Similarly, lines 333-335 state that "By reconstructing the evolutionary history of signal's information content in parallel to signal structure, our work adds and quantifies a functional perspective to evolutionary patterns, thereby offering novel insights into animal signal evolution." Again, the implication is that a strong function of drumming structure is inter-specific recognition. But then on line 350, they state that "...our results suggest that woodpeckers' drumming signal has not been selected for maximizing information for species identity in the signal itself but that it has nonetheless preserved the efficacy of its species'

signature as the number of species in the clade increased.” Then on line 363 the implication is that drumming patterns are unrelated to mating behavior and sexual selection.

In short: I still like this paper. I think that it could provide a very important example of the study of signal evolution in a system where the signal is simple but accessible to a broad experimental approach. However, from my perspective, the logic of the argument still needs to be cleaned up. **We fully understand the reviewer’s point and agree. By no means we imply that sexual selection should be neglected when considering signal evolution. On the contrary, we believe that selection on signals both between and within species are intertwined: in the same way that sexual selection occurs through choices from conspecifics, it is also conspecifics that apply a selective pressure to discriminate conspecific signals from heterospecific ones. In other words, given, e.g., two closely related species, intra-specific selection will also directly inter-specific divergence. In this sense, the wording ‘inter-specific selection’ does not seem appropriate here.**

In the present study we focus on one type (among many) of information that can be encoded in an acoustic signal, i.e. species identity. We could have similarly investigated information encoding e.g. body size (as often used to investigate the effect of sexual selection on acoustic signal structure), but the main goal here was not to identify which aspects of drumming could encode information about the quality of a potential partner/mate.

In line with the above and in order to not mislead our readers, we have modified our text, making it clear that, while recognizing other types of information, here we focus on information about species identity. In addition, we have attempted to correct the impression of ambiguity regarding the back-and-forth between species identity and sexual selection (note that, again, we do not necessarily see those as fully independent) and would like to thank the reviewer for pointing this out.

Edits:

Introduction, lines 41-43: “Signals can encode various types of information, including static (e.g. body size, sex, age, identity) and dynamic (e.g. arousal level, physiological states) attributes of the emitter, and are often subject to both sexual and natural selection pressures.”

Discussion, lines 390-394: “A possible outcome of reducing ecological interactions between different species (in this case foraging overlap) is that species recognition based on drumming patterns may be an epiphenomenon of factors unrelated to mating behaviour or sexual selection per se. Indeed, sexual selection may only mildly contribute to shaping drumming structure, and instead could involve multiple signaling modes including e.g. color or movement displays”

Minor points:

Line 35: make the sentence more explicit. “... species recognition in species assemblages that lack closely related species with concomitantly similar drum patterns”.

We have modified that sentence in the abstract. It now reads (lines 31-33): “Playback experiments and quantification of species discriminability demonstrate sufficient signal differentiation to support species recognition in local communities”

Line 38: Would this work better? "... the effectiveness of information transfer relevant to inter-specific discrimination".

Yes, thanks. We have modified the text accordingly.

Line 298: "Through this process, we could expect phenotypic (in this case, drumming structure) differences to be as high for closely related species that share the same geographical area as for distantly related species or species that do not co-exist." "... or species that do not co-exist"? This needs to be clearer.

Thank you for pointing this out. We have modified this sentence, which now reads '...or allopatric species'.

Line 370: Should "underestimate" be "overestimate"?

We actually did mean 'underestimate', in the sense that selection pressure to increase species-specific information could be stronger than we conclude, but overshadowed by mechanical constraints. We have slightly modified the text to make it more obvious, and it now reads (lines 400-402) 'In this higher selection pressure scenario, woodpecker species would simply not be able to respond to such a high selection pressure, and thus remain constrained in their ability to evolve their signals.'

1034: "Tukey"?

Yes. Thanks for spotting this.